



# 1 NAQPMS-PDAF v2.0: A Novel Hybrid Nonlinear Data Assimilation
# 2 System for Improved Simulation of PM₂.₅ Chemical Components

Hongyi Li[1,3], Ting Yang[1], Lars Nerger[4], Dawei Zhang[2], Di Zhang[2], Guigang Tang[2], Haibo Wang[1], Yele
Sun[1,3], Pingqing Fu[5], Hang Su[1,6], Zifa Wang[1,3]
[1]State Key Laboratory of Atmospheric Boundary Layer Physics and Atmospheric Chemistry (LAPC), Institute of Atmospheric
Physics, Chinese Academy of Sciences, Beijing 100029, China.
[2]China National Environmental Monitoring Centre, Beijing, China
[3]College of Earth and Planetary Sciences, University of Chinese Academy of Sciences, Beijing 100049, China
[4]Alfred Wegener Institute, Helmholtz Center for Polar und Marine Research, Bremerhaven, Germany
[5]Institute of Surface-Earth System Science, School of Earth System Science, Tianjin University, Tianjin 300072, China
[6]Max Planck Institute for Chemistry, Mainz, Germany
*Correspondence to*: Ting Yang (tingyang@mail.iap.ac.cn)
**Abstract.** PM₂.₅, a complex mixture with diverse chemical components, exerts significant impacts on the environment, human
health, and climate change. However, precisely describing spatiotemporal variations of PM₂.₅ chemical components remains a
difficulty. In our earlier work, we developed an aerosol extinction coefficient data assimilation (DA) system (NAQPMS-PDAF
v1.0) that is suboptimal for chemical components. This paper introduces a novel hybrid nonlinear chemical DA system
(NAQPMS-PDAF v2.0) to accurately interpret key chemical components ($SO_4^{2-}$, $NO_3^-$, $NH_4^+$, OC, and EC). NAQPMS-PDAF
v2.0 improves upon v1.0 by effectively handing and balancing stability and nonlinearity in chemical DA, which is achieved
by incorporating the non-Gaussian-distribution ensemble perturbation and hybrid Localized Kalman-Nonlinear Ensemble
Transform Filter with an adaptive forgetting factor for the first time. The dependence tests demonstrate that NAQPMS-PDAF
v2.0 provides excellent DA results with a minimal ensemble size of 10, surpassing previous reports and v1.0. A one-month
DA experiment shows that the analysis field generated by NAQPMS-PDAF v2.0 is in good agreement with observations,
especially reducing the underestimation of $NH_4^+$ and $NO_3^-$ and the overestimation of $SO_4^{2-}$, OC, and EC. In particular, the
CORR values for $NO_3^-$, OC, and EC are above 0.96, and $R^2$ values are above 0.93. NAQPMS-PDAF v2.0 also demonstrates
superior spatiotemporal interpretation, with most DA sites showing improvements of over 50%-200% in CORR and over 50%-
90% in RMSE for the five chemical components. Compared to the poor performance in global reanalysis dataset (CORR:
0.42-0.55, RMSE: 4.51-12.27 µg/m³) and NAQPMS-PDAF v1.0 (CORR: 0.35-0.98, RMSE: 2.46-15.50 µg/m³), NAQPMS-
PDAF v2.0 has the highest CORR of 0.86-0.99 and the lowest RMSE of 0.14-3.18 µg/m³. The uncertainties in ensemble DA
are also examined, further highlighting the potential of NAQPMS-PDAF v2.0 for advancing aerosol chemical component
studies.



## 1 Introduction

PM$_{2.5}$ is a complex mixture of various chemical fractions, mainly including sulfate (SO$_4^{2-}$), nitrate (NO$_3^-$), ammonium (NH$_4^+$),
organic carbon (OC), and elemental carbon (EC), which diversely influences the atmospheric environment (Khanna et al.,
2018), human health (Bell et al., 2007; Schlesinger, 2007; Li et al., 2022a; Alves et al., 2023), and climate change (Schult et
al., 1997; Park et al., 2014; Wilcox et al., 2016). However, current detection technologies, such as field observation with in-
situ sampling and chemical analysis (Zhang et al., 2015; Ming et al., 2017), remote-sensing inversion (Nishizawa et al., 2008;
Nishizawa et al., 2011; Nishizawa et al., 2017), and machine learning (Lin et al., 2022; Su Lee et al., 2023) are insufficient in
interpreting PM$_{2.5}$ chemical components due to the spatiotemporal discontinuity and limited chemical species. Although
atmospheric chemistry transport models (CTMs) (Wang et al., 2014; Wang et al., 2015; Jia et al., 2017; Yang et al., 2019; Li
et al., 2020; Lv et al., 2020) are commonly used to characterize spatiotemporal distribution of multiple chemical species, CTMs
are associated with uncertainties in initial-boundary conditions, physiochemical mechanisms, emission inventories, and
meteorological fields (Sax and Isakov, 2003; Mallet and Sportisse, 2006; Rodriguez et al., 2007; Chang et al., 2015; Miao et
al., 2020; Xie et al., 2022), resulting in biases relative to real situation.

Data assimilation (DA) offers a solution to integrate the multi-source observations, CTMs, and their uncertainties effectively
to enhance the simulation and forecasting capabilities of CTMs. Variational methods (3D-Var/4D-Var) (Talagrand and Courtier,
1987), Ensemble Kalman Filter (EnKF) (Evensen, 1994; Evensen, 2003), EnKF-variants (EnKFs) (Bishop et al., 2001; Tippett
et al., 2003; Hunt et al., 2007; Nerger et al., 2012), and hybrid EnKF-Var methods (Hamill and Snyder, 2000; Schwartz et al.,
2014) are most widely applied in DA. However, variational methods have a flow-independent Background Error Covariance
(BEC) with the assumption of isotropic, static, and uniform characteristics, and they need to develop the tangent linear adjoint
model, which is difficult to practice for complex models. Although EnKFs and hybrid EnKF-Var methods have a flow-
dependent BEC, they are sensitive to inadequate ensemble sampling and have high computational costs. Importantly, these
methods cannot address model nonlinearity and non-Gaussian error distribution, yielding suboptimal results for DA in highly
nonlinear CTMs.

Currently, nonlinear filters, such as Particle Filter (PF) (Gordon et al., 1993) and Nonlinear Ensemble Transform Filter (NETF)
(Tödter and Ahrens, 2015), have been proposed to approximate the complete posterior probability distribution of model states
and provide a better representation of non-Gaussian information based on Monte Carlo random sampling and Bayesian theory.
However, PF is unstable and susceptible to filter degeneration compared to EnKFs. In a recent study, Nerger (2022) proposed
the hybrid Kalman-Nonlinear Ensemble Transform Filter (KNETF) to achieve excellent DA performance in the Lorenz-63 and
Lorenz-96 model with a smaller ensemble size, which combines the stability of EnKFs and the nonlinearity of NETF (Nerger,

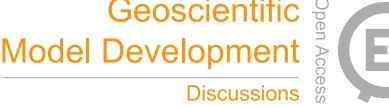

2022). However, to the author's knowledge, this algorithm has not been applied to the chemical DA of CTMs.

Studies on chemical DA involve the assimilation of aerosol optical properties, such as aerosol optical depth (AOD) and
extinction coefficient (EXT), and the particulate matters (PMs), such as the mass concentrations of $PM_{2.5}$ and $PM_{10}$. The
commonly AOD observations for DA include OMI-AOD (Ali et al., 2013), MODIS-AOD (Zhang et al., 2008; Huneeus et al.,
2012; Huneeus et al., 2013; Rubin and Collins, 2014; Lynch et al., 2016; Werner et al., 2019; Kumar et al., 2020), AERONET-
AOD (Schutgens et al., 2010; Li et al., 2016), Sun-Sky Photometer-Multiband AOD (Chang et al., 2021), GOCI-AOD (Saide
et al., 2014; Luo et al., 2020; Kim et al., 2021), and Fengyun/Himawari8-AOD (Bao et al., 2019; Jin et al., 2019; Xia et al.,
2019; Xia et al., 2020). These studies indicated that AOD observations can enhance the accuracy of aerosol simulation and
forecast. Compared to AOD, EXT DA effectively improves the interpretation of aerosol vertical distribution (Zhang et al.,
2014; Cheng et al., 2019; Wang et al., 2022). Additionally, the simultaneous DA of aerosol optical properties and PMs is widely
applied in aerosol studies (Tang et al., 2015; Chai et al., 2017). According to our literature review (Yang et al., 2023), there is
currently no DA study on aerosol chemical components due to the limited DA influence of PMs and AOD on chemical
compositions (Chang et al., 2021) and the limited chemical observations with an extensive spatial range. Moreover, the aerosol
chemical components exhibit nonlinearity and a non-Gaussian distribution (Ha, 2022), while current main-stream algorithms,
such as variational methods or EnKFs, are suboptimal for chemical component DA.

In our previous work, we developed an aerosol vertical DA system (NAQPMS-PDAF v1.0) based on EnKFs to improve the
simulation of the extinction coefficient vertical profile (Wang et al., 2022). In this study, we present a novel hybrid nonlinear
DA system (NAQPMS-PDAF v2.0) towards various $PM_{2.5}$ chemical components through online integration of Parallel Data
Assimilation Framework (PDAF, version 2.1, released on February 21[st], 2023), Observation Module Infrastructure (OMI) and
Nested Air Quality Prediction Model System (NAQPMS). We collected 1-month hourly surface observations of five $PM_{2.5}$
chemical components ($NH_4^+$, $SO_4^{2-}$, $NO_3^-$, OC, and EC) over Northern China and surrounding areas. We utilized the hybrid
Localized Kalman-Nonlinear Ensemble Transform Filter (LKNETF) to generate a high-resolution and high-accuracy
reanalysis dataset of $PM_{2.5}$ chemical components for the first time. Notably, the ensemble members in NAQPMS-PDAF v2.0
are generated by perturbing emission species based on their uncertainties and non-Gaussian distribution assumption. Section
2 briefly introduces NAQPMS and PDAF v2.1 with OMI, respectively, and details the development of NAQPMS-PDAF v2.0,
including system structure, configuration, ensemble generation, and LKNETF algorithm. The data used in this study and
experimental settings are also described in Section 2. Section 3 presents the DA results, including evaluating dependencies,
performance, and external comparisons. Besides, Section 3 discusses the ensemble DA uncertainty. Section 4 summarizes the
conclusions and outlook.



**2 Method and data**
**2.1 NAQPMS**
The Nested Air Quality Prediction Modeling System (NAQPMS), developed by the Institute of Atmospheric Physics
(IAP), Chinese Academy of Sciences (CAS), is used to provide background fields of key aerosol chemical components in this
study. NAQPMS is capable of characterizing the three-dimensional spatiotemporal distribution of various atmospheric
compositions at global and regional scales through multiple physicochemical processes (shown in Table S1) and has been
widely used in atmospheric pollution and chemistry research, such as $O_3$ pollution, haze episodes (Wang et al., 2014; Du et al.,
2021), regional transport (Wang et al., 2017; Wang et al., 2019), source identification (Li et al., 2022b), air quality simulation
at global scale (Ye et al., 2021) and at urban-street scale (Wang et al., 2023), and acid deposition (Ge et al., 2014).
**2.2 PDAF v2.1 with OMI**
The Parallel Data Assimilation Framework (PDAF, https://pdaf.awi.de/trac/wiki) is an open-source and high-expandability
software developed by the Alfred Wegener Institute (AWI) in Germany to integrate observations, numerical models, and
assimilation systems for DA tasks, widely applied in meteorology, oceanography, land surface and atmospheric chemistry
(Kurtz et al., 2016; Nerger et al., 2020; Mingari et al., 2022; Strebel et al., 2022; Wang et al., 2022; Yu et al., 2022). The initial
version of PDAF (PDAF v1.0) was released in 2004. It has undergone continuous improvements and updates, with major
updates including the introduction of Ensemble Transform Kalman Filter (ETKF) and its localized variant (LETKF) in version
1.6, the implementation of PDAF-OMI (Observation Module Infrastructure) in version 1.16, the integration of 3D-Var methods
in version 2.0, and the incorporation of the hybrid KNETF and its localized variant (LKNETF) for the first time in version 2.1,
which was released in 2023 to handle the complex DA situations, such as the nonlinearity of system and non-Gaussian error
distribution of model state. Notably, the version of PDAF coupled in NAQPMS-PDAF v1.0 is PDAF v1.15 (released in 2019),
implying that NAQPMS-PDAF v1.0 has more limited applicability and functionality. In this work, the PDAF v2.1 is coupled
in NAQPMS-PDAF v2.0.

PDAF has two modes, namely offline and online mode. For the offline mode, PDAF and the model perform separately without
coupling, which is easy to write code. For the online mode, PDAF is coupled with the model, and model calculation and data
assimilation perform continuously. Compared to the offline mode, the online coupling has several advantages. Firstly, the
initialization process of PDAF and the model only needs to be executed once instead of twice independently. Secondly, the
model integration result can be directly passed to PDAF for data assimilation. Additionally, the assimilation result of PDAF
can be directly passed to the model for the next model integration. This eliminates the need for intermediate steps and improves
efficiency. Thirdly, the online mode is controlled by a main program, which allows for efficient use of several processors in





the high-performance computing cluster. Conversely, in the offline mode, the PDAF and the model are managed by distinct
programs, often with a reduced number of processors available for each program. Therefore, the online-mode PDAF is used
in this study.

PDAF-OMI, an extension of PDAF, provides I/O interfaces for multi-type observations, simplifying user observation handling
by offering generic PDAF-OMI core routines and independent user-supplied routines for each observational type. The user-
supplied routines, namely *init_dim_obs/init_dim_obs_l*, *obs_op*, and *localize_covar*, are responsible for reading and writing
multi-type observations, applying corresponding observation operators, and performing covariance localization, respectively.
The modules for all observation types are integrated into the *callback_obs_pdafomi*, allowing free combinations between
different observation types without interference and facilitating the collaborative DA for various aerosol chemical components.
PDAF-OMI was not applied in NAQPMS-PDAF v1.0. Consequently, NAQPMS-PDAF v1.0 cannot switch between different
observational type combinations, and users need to define complete routines for each observation type for the DA process,
resulting in more tedious code writing and higher computational costs in NAQPMS-PDAF v1.0.
**2.3 NAQPMS-PDAF v2.0**
**2.3.1 Structure of NAQPMS-PDAF v2.0**
Figure 1 illustrates the structure and main workflow of NAQPMS-PDAF v2.0. The observational part involves multi-type
observations and PDAF-OMI. PDAF-OMI enables the simultaneous access and scheduling of multi-type and multi-source
data through observational indices, which allows for flexible combination. The ensemble forecast/background fields are
generated by perturbing emission species (see Sect. 2.3.3) and NAQPMS calculations (the green part in Fig. 1). Then chemical
DA is performed by a novel hybrid localized nonlinear DA algorithm (LKNETF, see Sect. 2.3.4) with an adaptive hybrid
weight and an adaptive forgetting factor to generate analysis/initial fields for the next realization.



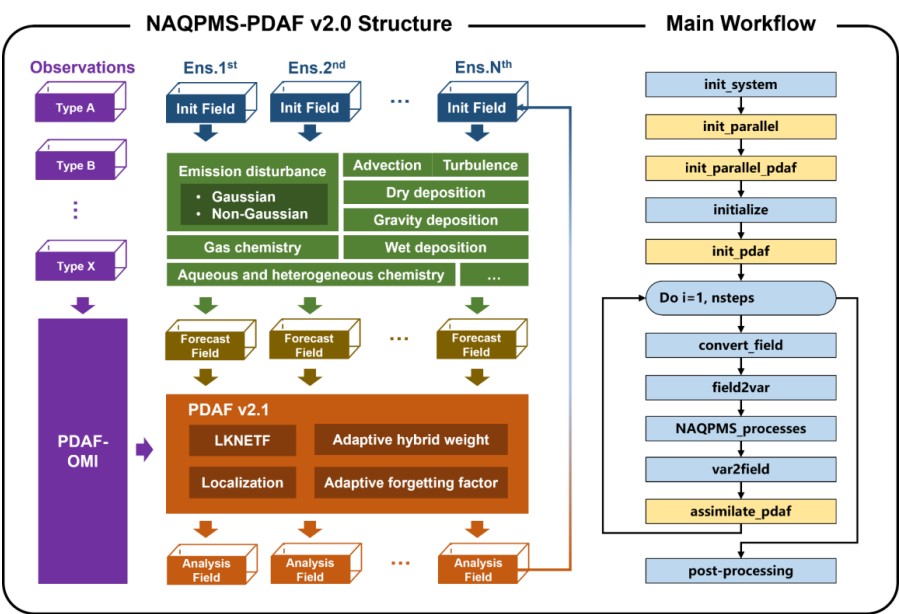

**Figure 1: The structure of NAQPMS-PDAF v2.0**

NAQPMS-PDAF v2.0 implements an online coupling between NAQPMS and PDAF v2.1 with OMI, utilizing a level-2 parallel computational framework. The online coupling ensures the continuous operation of model forecasts and assimilation analysis at each time step, achieved by directly integrating PDAF routines into the prototype code of NAQPMS (In Fig. 1 right part, the blue represents NAQPMS main routines, while the yellow represents PDAF main routines). The level-2 parallel computational framework, which utilizes the Message Passing Interface standard (MPI), facilitates concurrent processing and data exchange among multiple ensemble members and parallel computation among model state matrixes within each ensemble member, enhancing the efficiency of ensemble analysis and numerical model computations. The description of level-2 parallel implementation was detailed in our previous work (Wang et al. 2022). The workflow of NAQPMS-PDAF v2.0 is outlined as follows:

Step 1. *init_system* module initializes NAQPMS, such as defining all model state variables, allocating numerical matrixes, configuring parameters, I/O of meteorological fields, and emission input.

Step 2. *init_parallel* module initializes MPI (MPI_COMM_WORLD) and model communicator (MPI_COMM_MODEL), their number of processes, and the rank of a process, followed by *init_parallel_pdaf*, which initializes MPI communicators for the model tasks, filter tasks and the coupling between model and filter tasks.

Step 3. *initialize* module initializes the target field (such as $PM_{2.5}$ chemical components), such as their spatiotemporal dimensions (longitude, latitude, and time steps) and variable dimensions.

Step 4. *init_pdaf* module initializes PDAF variables, such as the local state dimension, global state dimension, and settings for analysis steps.





Step 5. Perform the time loop of forecast and analysis. The *convert_field* module is employed to match the matrix storage rule
of the target field between NAQPMS and PDAF to ensure compatibility. The *field2var* module collects the analysis field/initial
field and establishes a relationship between the initial field/analysis field and sub-variables in NAQPMS. Subsequently, the
analysis field values are allocated to the corresponding NAQPMS sub-variables, and then the *NAQPMS_processes* module
performs the forecast. After that, the *var2field* module, the inverse of the *field2var* module, assigns the NAQPMS sub-variables
to the forecast field/background field. Finally, the *assimilate_pdaf* module assimilates the target field with observations to
generate an analysis field for the next iteration.
Step 6. post-processing is responsible for finalizing NAPQMS-PDAF, data analysis, and DA evaluation.
**2.3.2 Configures**
The meteorological field for NAQPMS is provided by the Weather Research and Forecasting model version 4.0 (WRFV4.0,
https://www.mmm.ucar.edu/models/wrf). The initial-boundary conditions for WRF are obtained from NCEP GDAS Final
Analysis (https://rda.ucar.edu/datasets/ds083.3/), with a horizontal resolution of 0.25°×0.25° and the temporal resolution of 6
hours, produced by the Global Data Assimilation System (GDAS). The land use data for WRF was updated by USGS's
MCD12Q1 v006 in 2019 (https://lpdaac.usgs.gov/products/mcd12q1v006/) with 20 categories. Three nested model domains
are conducted with the horizontal resolutions of 45 km in the East Asia region (domain1), 15 km in most areas of China except
for the western area (domain2), and 5 km in the Northern China region (domain3, target research region). WRF and NAQPMS
have 40 vertical layers with 27 layers within 2 km. The parameterization schemes for physical processes in WRF are shown
in Table S2. The boundary condition input for NAQPMS is provided by the global chemistry transport Model for OZone And
Related chemical Tracers version 2.4 (MOZART V2.4) (Horowitz et al., 2003). The anthropogenic emissions for NAQPMS
are from Tsinghua University's 2016 Multi-resolution Emissions Inventory for China (MEIC, http://www.meicmodel.org/)
with a spatial resolution of 0.25°×0.25°, including residential sources, transportation sources, agricultural sources, industrial
sources, and power plant sources. The computational platform is the high-performance supercomputer subsystem cluster with
320 computation nodes, a total of 12,800 processors, and about 153 TB memory at the Big Data Cloud Service Infrastructure
Platform (BDCSIP), which meets the demand for high-performance parallel computing of NAQPMS-PDAF v2.0.

The model state variables include $NH_4^+$, $SO_4^{2-}$, $NO_3^-$, OC, EC, $Na^+$, Brown carbon, soil $PM_{2.5}$, soil $PM_{10}$, sea salt, fine dust,
coarse dust, $SO_2$, $NO_2$ and RH. As shown in Fig. 2, the model state has a 4-dimensional (4-D) structure, with longitudinal
dimension (ix, 300 grids), latitudinal dimension (iy, 249 grids), variable dimension (ivar, 15), and vertical dimension (iz, 40
layers) in that order. The 4-D model state with 15 variables is converted to a 2-D state matrix in PDAF, the number of grids in
the horizontal axis direction is ix, and the number of grids in the vertical axis direction is iy*ivar*iz. Notably, the coordinate
index of the 2-D state matrix contains 3-D information for each variable to implement the horizontal and vertical domain



localization separately, because the horizontal and vertical resolutions are not uniform. This structure has two advantages. First,
the parallel cutting of the horizontal axis enables the local domain to retain the full dimensional information (ix_p*iy*ivar*iz,
where ix_p is the longitudinal dimension of the local domain). Secondly, the localization in local domain permits the analysis
only executes within a small domain (ix_p*iy) when the length of horizontal localization radius (Rs) is smaller than iy, which
effectively reduces the influence of spurious correlations between different state variables. In this study, we set the horizontal
and vertical domain localization radius to 200 km (40 grids) and 1 layer. Besides, we further implemented the observation
localization to consider the influence of distance between analysis grid and observational grid (see Sect. 2.3.4). To minimize
computational complexity, the observation errors were assumed to be spatially isotropic, with 0.40 μg/m$^3$, 1.00 μg/m$^3$, 0.50
μg/m$^3$, 3.00 μg/m$^3$, and 0.50 μg/m$^3$ for $NH_4^+$, $SO_4^{2-}$, $NO_3^-$, OC and EC, respectively.

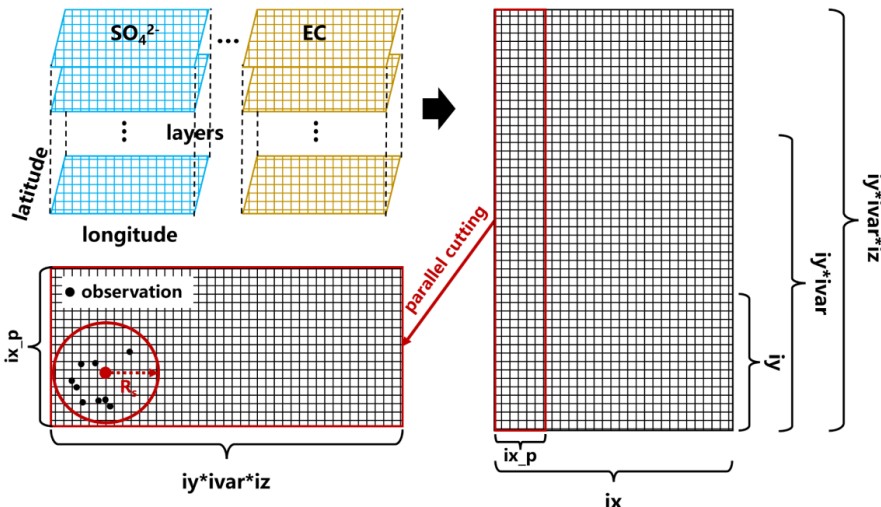


**Figure 2: The structure of state variables in NAQPMS-PDAF v2.0.**
**2.3.3 Generation of ensemble members**
In ensemble DA, ensemble members interpret the uncertainty of the model or system, characterized by BEC, which
significantly impacts the DA performance (Dai et al., 2014). For CTMs, emission input directly influences the chemical
calculation and substantially contributes to the uncertainty. Perturbing emission input can effectively represent the uncertainty
in aerosol emissions and enhance the consistency of ensemble error spread, thereby improving aerosol DA (Huang et al., 2023).
CTMs are nonlinear, and model state errors are non-Gaussian distributions. To obtain non-Gaussian error distributions, we
followed Kong et al. (2021)'s method to assume that the emission errors are spatially correlated by an isotropic correlation
model with the decorrelation length of 150 km and generate perturbation coefficient matrixes with the same Gaussian
distribution as the emission species, which are subsequently transformed into non-Gaussian distribution matrixes through non-
Gaussian process generation v1.2 (https://github.com/ECheynet/Gaussian_to_nonGaussian/).




The target $PM_{2.5}$ chemical components are $NH_4^+$, $SO_4^{2-}$, $NO_3^-$, OC, and EC, and the perturbed emission species correspondingly
include $SO_2$, NOx, VOCs, $NH_3$, CO, $PM_{10}$, $PM_{2.5}$, EC, and OC, with the corresponding uncertainties ($\delta$) listed in Table 1. As
shown in Eq. (1), the original emission input matrix ($E_p$) is multiplied by the corresponding perturbation coefficient matrix
($\theta_i$) to generate the perturbed emission input matrix ($E_i$) for each emission species. The calculation of the perturbation
coefficient matrix ($\theta_i$) is followed by Eq. (2)-(3). Firstly, N two-dimensional pseudorandom perturbation fields ($P_i$) are created
using Evensen's method (Evensen, 1994). The uncertainties ($\delta$) of the emission species are incorporated into the two-
dimensional pseudorandom perturbation fields ($P_i$) to obtain the final perturbation coefficient matrixes ($\theta_i$). Finally, the
Gaussian-distribution perturbation coefficient matrixes ($\theta_i$) were transformed into non-Gaussian distribution coefficient
matrixes with a given target skewness (set to 1) and kurtosis (set to 6) by non-Gaussian process generation v1.2, which employs
the Moment Based Hermite Transformation Model and a cubic transformation.
**Table 1: The uncertainties of emission species in NAQPMS-PDAF v2.0**

| Species | $SO_2$ | NOx | VOCs | $NH_3$ | CO | $PM_{10}$ | $PM_{2.5}$ | EC | OC |
|---|---|---|---|---|---|---|---|---|---|
| Uncertainty $\delta$ | 2.00 | 0.31 | 0.68 | 0.53 | 0.70 | 1.32 | 1.30 | 2.08 | 2.58 |

$$E_i = E_p \times \theta_i, i = 1,2, \dots, N \ , \tag{1}$$
$$\ln \theta_{o_i} = (\frac{(P_i - \frac{1}{N} \times \sum_{i=1}^N P_i)}{\sqrt{\frac{1}{N} \times \sum_{i=1}^N (P_i - \frac{1}{N} \times \sum_{i=1}^N P_i)^2}} - \frac{1}{2} \times \ln(1+\delta^2)) \times \sqrt{\ln(1+\delta^2)} \ , \tag{2}$$
$$\theta_i = \frac{\left(\theta_{o_i} - \frac{1}{N} \times \sum_{i=1}^N \theta_{o_i}\right)}{\sqrt{\frac{1}{N} \times \sum_{i=1}^N \left(\theta_{o_i} - \frac{1}{N} \times \sum_{i=1}^N \theta_{o_i}\right)^2}} \times \left(\frac{1}{N} \times \sum_{i=1}^N \theta_{o_i}\right) \times \delta + \frac{1}{N} \times \sum_{i=1}^N \theta_{o_i} \ , \tag{3}$$
Notably, all matrix operations involved are Schur Product. Where $E_i$ denotes the $i^{th}$ ensemble perturbed emission input matrix,
$E_p$ denotes the original unperturbed emission input matrix and $\theta_i$ represents the $i^{th}$ ensemble perturbation coefficient matrix.
$\theta_{o_i}$ is the $i^{th}$ ensemble original perturbation coefficient matrix, which is obtained by mathematical transformation of the $i^{th}$
ensemble pseudorandom perturbation matrix $P_i$, including standardization, scaling by uncertainty ($\delta$), and logarithm.
**2.3.4 Hybrid nonlinear DA algorithm with adaptive forgetting factor**
To thoroughly integrate the stability of EnKFs with the nonlinearity of nonlinear filters and be ideal for the nonlinear and non-
Gaussian-distribution situations, the hybrid LKNETF is used in this study. This section reviews the algorithms of LETKF,
LNETF, and their combination (LKNETF).

ETKF, a deterministic filter in EnKFs, efficiently obtains analysis samples using a transformation matrix and the square root
of the forecast error covariance (Bishop et al., 2001). In contrast to stochastic filters in EnKFs, ETKF prevents underestimation
of the analysis error covariance resulting from the random observation perturbations. And it is particularly applicable in
situations with small ensemble sizes (Lawson and Hansen, 2004). The realization of ETKF can be divided into the forecast



and analysis steps.

In the forecast step, the forecast state vector ($\mathbf{x}_t^f$) at t is generated by numerical model ($\mathbf{M}$) integration of the analysis state
vector ($\mathbf{x}_{t-1}^a$) at t-1. The forecast error covariance matrix ($\mathbf{P}_t^f$) can be calculated by the perturbation of the forecast ensemble
($\mathbf{X}_t^{f'}$).
$$\mathbf{x}_t^f = \mathbf{M}(\mathbf{x}_{t-1}^a), \mathbf{X}_t^f = [\mathbf{x}_{1_t}^f, \mathbf{x}_{2_t}^f, \dots, \mathbf{x}_{K_t}^f] \ , \qquad (4)$$
$$\mathbf{P}_t^f = \mathbf{X}_t^{f'} \mathbf{X}_t^{f'^T} \ , \qquad (5)$$
Where $\mathbf{X}_t^f$ is the forecast ensemble at t, and K is the number of ensemble members. $\mathbf{X}_t^{f'}$ is the perturbation of the forecast
ensemble at t, calculated by $\mathbf{X}_t^f$ and the forecast ensemble mean $\overline{\mathbf{X}_t^f}$ at t.

In the analysis step, the forecast error covariance matrix ($\mathbf{P}_t^f$) at t is transformed to the analysis error covariance matrix ($\mathbf{P}_t^a$) at
t by a transform matrix ($\mathbf{T}$).
$$\mathbf{P}_t^a = \mathbf{X}_t^{f'} \mathbf{T} \mathbf{X}_t^{f'^T} \ , \qquad (6)$$
The transform matrix ($\mathbf{T}$) is defined as follows and can be decomposed to a left singular vector matrix ($\mathbf{U}$), a singular value
matrix ($\mathbf{S}$), and a right singular vector matrix ($\mathbf{V}$) through the singular value decomposition.
$$\mathbf{T}^{-1} = \rho_{adaptive}(K-1)\mathbf{I} + (\mathbf{HX}_t^{f'})^T(\mathbf{L} \cdot \mathbf{R}^{-1})\mathbf{HX}_t^{f'} = \mathbf{USV} \ , \qquad (7)$$
$$\rho_{adaptive} = \frac{\sigma_{ens}^2}{\sigma_{resid}^2 - \sigma_{obs}^2} \ , \qquad (8)$$
Where $\rho_{adaptive}$ is an adaptive forgetting factor, used for the inflation of error covariance estimation (the initial $\rho_{adaptive}$ is
set to 0.9 in this study). $\sigma_{ens}^2$ is the mean ensemble variance, $\sigma_{resid}^2$ is mean of observation-minus-forecast residual, $\sigma_{obs}^2$ is
mean observation variance. $\mathbf{I}$ is the identity matrix. $\mathbf{H}$ is the observation operator. $\mathbf{L}$ is the localization matrix, a weight
matrix calculated by the 5[th]-order polynomial (Nerger, 2015), implemented in LETKF for observation localization analysis to
avoid observational spurious correlation and filter divergence effectively (Hunt et al., 2007). $\mathbf{R}$ is the observation error
covariance matrix.

The analysis state vector ($\mathbf{x}_t^a$) at t is calculated by the forecast state vector ($\mathbf{x}_t^f$) at t, the perturbation of the forecast ensemble
($\mathbf{X}_t^{f'}$) at t and a weight vector ($\mathbf{w}$).
$$\mathbf{x}_t^a = \mathbf{x}_t^f + \mathbf{X}_t^{f'} \mathbf{w} \ , \qquad (9)$$
The weight vector ($\mathbf{w}$) is given by the following equation.





$\quad \mathbf{w} = \mathbf{T}(\mathbf{HX}_t^{f'})^T (\mathbf{L} \cdot \mathbf{R}^{-1})(\mathbf{y} - \mathbf{Hx}_t^f)$ ,  $\hfill$ (10)
$\quad$ Where $\mathbf{y}$ is observations.

$\quad$ The analysis ensemble ($\mathbf{X}_t^a$) at t can be obtained by forecast ensemble mean ($\overline{\mathbf{X}_t^f}$) at t, the perturbation of the forecast ensemble
$\quad$ ($\mathbf{X}_t^{f'}$) at t and a transform matrix ($\mathbf{C}$) represented by the symmetric square root of $\mathbf{T}$.
$\quad \mathbf{X}_t^a = \overline{\mathbf{X}_t^f} + \sqrt{K-1}\,\mathbf{X}_t^{f'}\mathbf{C}$ ,  $\hfill$ (11)
$\quad$ The transform matrix ($\mathbf{C}$) is calculated as follows.
$\quad \mathbf{C} = \mathbf{US}^{-1/2}\mathbf{U}^T$ ,  $\hfill$ (12)
$\quad$ NETF is a 2$^{nd}$-order exact ensemble square root filter effectively applied to the nonlinear and non-Gaussian DA (Tödter and
$\quad$ Ahrens, 2015). Like PF, NETF indirectly updates the model state by using observations to affect the weights of the prior
$\quad$ ensemble. However, PF and NETF differ in the sampling method. PF utilizes the Monte Carlo and Bayesian methods to
$\quad$ calculate particle weights based on observations, which are then used to generate the analysis ensemble by weighting the
$\quad$ resampling forecast ensemble. In high-dimensional systems, as the DA progresses, the weight differences of particles increase,
$\quad$ with most particles having weights close to 0, leading to filter degeneration. In contrast, NETF generates the analysis ensemble
$\quad$ through a deterministic matrix square root transformation of the forecast ensemble, where the mean and covariance matrix of
$\quad$ the analysis ensemble match the weighted values in PF (as shown in Eq. (13)-(14)). Due to the similarity between NETF and
$\quad$ ETKF, the localization can be implemented in NETF (LNETF) (Tödter et al., 2016).
$\quad \bar{\mathbf{x}}^a = \frac{1}{K}\sum_{i=1}^{K}\mathbf{x}_i{}^a = \frac{1}{K}\sum_{i=1}^{K}w_i\mathbf{x}_i{}^f$ ,  $\hfill$ (13)
$\quad$ Where $\bar{\mathbf{x}}^a$ is the analysis state vector mean, K is the number of ensemble members, $\mathbf{x}_i{}^a$ is the i$^{th}$ analysis state vector, $w_i$ is
$\quad$ the i$^{th}$ particle weight vector in PF, which is calculated by the Bayesian method $w_i = p(\mathbf{y}|\mathbf{x}_i{}^f)/p(\mathbf{y})$, $\mathbf{y}$ is the observations,
$\quad \mathbf{x}_i{}^f$ is the i$^{th}$ forecast state vector.
$\quad \mathbf{P}^a = \frac{1}{K-1}\sum_{i=1}^{K}(\mathbf{x}_i{}^a - \bar{\mathbf{x}}^a)(\mathbf{x}_i{}^a - \bar{\mathbf{x}}^a)^T = \sum_{i=K}^{K}w_i(\mathbf{x}_i{}^f - \bar{\mathbf{x}}^f)(\mathbf{x}_i{}^f - \bar{\mathbf{x}}^f)^T$ ,  $\hfill$ (14)
$\quad$ Where $\mathbf{P}^a$ is the error covariance matrix of the analysis ensemble, calculated by the perturbation of the analysis ensemble.
$\quad$ In NETF, $\mathbf{A}$ performs as a transform matrix like the transform matrix ($\mathbf{T}$) in ETKF, which can be obtained from the weight
$\quad$ matrix ($\mathbf{w}$).
$\quad \mathbf{P}^a = \mathbf{X}^{f'}\mathbf{A}\mathbf{X}^{f'^T}$ ,  $\hfill$ (15)
$\quad \mathbf{A}^{1/2} = (\mathbf{W} - \mathbf{w}\mathbf{w}^T)^{1/2} = \mathbf{V}\mathbf{D}^{1/2}\mathbf{V}^T$ ,  $\hfill$ (16)



Where the matrix $\mathbf{W} \equiv \mathrm{diag}(\mathbf{w})$ is defined as a diagonal matrix created from the weight matrix ($\mathbf{w}$). $\mathbf{A}$ can be decomposed
($\mathbf{A} = \mathbf{VDV^T}$) by a singular value decomposition as it is a real, symmetric, positive semidefinite matrix. $\mathbf{V}$ is the orthogonal
matrix, and $\mathbf{D}$ is a diagonal matrix.

Then, the perturbation of the analysis ensemble ($\mathbf{X^{a'}}$) and the analysis ensemble ($\mathbf{X^a}$) can be obtained by applying the square
root of $\mathbf{A}$ as a transform matrix.
$$\mathbf{X^{a'}} = \sqrt{K}\mathbf{X^{f'}}\mathbf{A}^{1/2} \ , \tag{17}$$
$$\mathbf{X^a} = \mathbf{\bar{X}^f} + \mathbf{X^{f'}}\left(\mathbf{\bar{W}} + \sqrt{K}\mathbf{A}^{1/2}\right) \ , \tag{18}$$
LKNETF combines the LETKF and LNETF through a hybrid weight $\gamma$ to perform better in systems with different non-
linearity degrees and implement in situations with smaller ensemble sizes (Nerger, 2022). When $\gamma$ approaches 1, the analysis
increment ($\Delta\mathbf{X}_{LETKF}$) computed by LETKF becomes more significant and appropriate for linear systems with Gaussian
distributions. Conversely, when $\gamma$ approaches 0, the analysis increment ($\Delta\mathbf{X}_{LNETF}$) computed by LNETF becomes more
significant and appropriate for non-linear systems with non-Gaussian distributions. The one-step update scheme is used in this
study.
$$\mathbf{X}^a_{HSync} = \overline{\mathbf{X}^f} + (1 - \gamma)\Delta\mathbf{X}_{LNETF} + \gamma\Delta\mathbf{X}_{LETKF} \ , \tag{19}$$
**2.4 Data**
**2.4.1 Observation**
The one-month (February 2022) hourly mass concentration observations of five $PM_{2.5}$ chemical components ($NH_4^+$, $SO_4^{2-}$,
$NO_3^-$, OC, and EC) from 33 ground-based sites in Northern China and surrounding areas were collected for this work (Fig. 3).
Out of the 33 sites, 24 (DA sites) were utilized for DA and internal validation, and the remaining 9 (VE sites) were used for
independent verification to assess the influence of DA sites on neighboring areas. These sites were divided using the K-means
clustering algorithm (Lloyd, 1982; Arthur and Vassilvitskii, 2007). The supplement provides a detailed description (Text S1).
$PM_{2.5}$ hourly observations from the China National Environmental Monitoring Centre (CNEMC, http://www.cnemc.cn/) were
employed to assess the overall mass concentration of $PM_{2.5}$ chemical components in NAQPMS-PDAF v2.0. Due to incomplete
spatial overlap between the $PM_{2.5}$ sites and the chemical component sites, the $PM_{2.5}$ sites were selected based on the closest
coordinate Euclidean distance between $PM_{2.5}$ sites and chemical component sites.



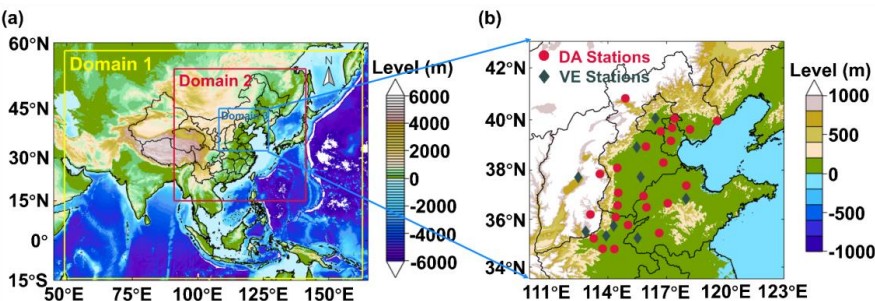

**Figure 3: The model domains in WRF simulation (a) and the location of observations (b). The domain 3 in (a) is the target area in this study. Twenty-four red sites in (b) represent the sites for data assimilation, and nine green sites in (b) represent the sites for spatial independent validation. The topographic dataset is from the ETOPO1 1 arc-minute Global Relief Model, taken from the National Geophysical Data Center (Amante and Eakins, 2009).**

### 2.4.2 Global reanalysis dataset

The global reanalysis datasets of $PM_{2.5}$ chemical components in February 2022 were obtained from the Copernicus Atmosphere Monitoring Service ReAnalysis (CAMSRA, 0.75°×0.75°) (Inness et al., 2019) and the Modern-Era Retrospective analysis for Research and Applications, Version 2 (MERRA-2, 0.5°×0.625°) (Randles et al., 2017) to compare with reanalysis dataset generated by NAQPMS-PDAF v2.0. For the consistency of data comparison, the global reanalysis surface grid data located in the observation sites of $PM_{2.5}$ chemical component were extracted through the k-nearest neighbor search method (Friedman et al., 1977), which can efficiently match grid points and observation sites based on longitude and latitude data and Euclidean distances. Our 3-hourly NAQPMS-PDAF v2.0 output of $NO_3^-$ and $NH_4^+$ were extracted to compare with the CAMSRA dataset, and hourly NAQPMS-PDAF v2.0 output of $SO_4^{2-}$, OC, and EC were extracted to compare with MERRA-2 M2T1NXAER dataset.

### 2.5 Experimental setting and evaluation method

In our study, four tests were conducted to evaluate the performance of NAQPMS-PDAF v2.0 with hourly observations of five $PM_{2.5}$ chemical components, including (1) the dependence on ensemble size and assimilation frequency, (2) the interpretation ability on mass concentration and spatiotemporal characteristics, (3) the superiority compared to other reanalysis dataset, and (4) the uncertainty in ensemble assimilation. In practice, the ratio of ensemble size to the number of processes with 1:50 in high-performance computers was the best parallel scheme to balance computing efficiency and computing resources (Wang et al., 2022).

All the tests were run in NAQPMS-PDAF v2.0 after a spin-up experiment with 24 timesteps from 00:00 to 23:00 (LST) on February 1st, 2022. (1) For the first test, we assimilated the hourly observations of five $PM_{2.5}$ chemical components from all sites with 48 timesteps from 00:00 (LST) on February 2nd to 23:00 (LST) on February 3rd, 2022. In the first situation, we





controlled a fixed assimilation frequency of 1 hour and changed the ensemble size to 2, 5, 10, 15, 20, 30, 40, and 50. In the
second situation, we controlled a fixed ensemble size of 20 and changed the assimilation frequency to 1 hour, 2 hours, 3 hours,
4 hours, 5 hours, 6 hours, 8 hours, and 12 hours. (2) For the second test, we set an ensemble size of 20 and an assimilation
frequency of 1 h and assimilated the hourly observations of five $PM_{2.5}$ chemical components from DA sites with 648 timesteps
from 00:00 (LST) on February $2^{nd}$ to 23:00 (LST) on February $28^{th}$, 2022. We also conducted a free running (FR) experiment
without assimilation in the same period for comparison. (3) For the third test, we followed the settings in the second test but
assimilated the observation from all sites to generate a high-quality reanalysis dataset of five $PM_{2.5}$ chemical components. (4)
The last test was like the first but with a different situation to investigate the impact of ensemble perturbation on ensemble
assimilation. From Table 2, we fixed species uncertainty (M4 setting) with five distribution types in the first situation and fixed
distribution type (T2 setting) with five $SO_2$ uncertainties in the second.
**Table 2: The experiment settings for emission perturbation**

| Experiment | Distribution (Fixed species uncertainty) |
|---|---|
| T1 | Gaussian |
| T2 | Non-Gaussian (m3=1, m4=6) |
| T3 | Non-Gaussian (m3=-1, m4=6) |
| T4 | Non-Gaussian (m3=1, m4=12) |
| T5 | Non-Gaussian (m3=-1, m4=12) |
| | $SO_2$ uncertainty (Fixed distribution) |
| M1 | 12% |
| M2 | 50% |
| M3 | 100% |
| M4 | 200% |
| M5 | 300% |


We used the Continuous Ranked Probability Score (CRPS) to evaluate ensemble size dependency, which measures the
consistency between ensemble forecast distribution and corresponding observations (Jolliffe and Stephenson, 2012). The
calculation rules are referred to in Hersbach's study (Hersbach, 2000). Besides, four common statistical indicators, the Pearson
correlation coefficient (CORR), root mean square error (RMSE), mean absolute error (MAE), and coefficient of determination
($R^2$), were used to assess the DA system performance in interpreting $PM_{2.5}$ chemical components ($SO_4^{2-}$, $NO_3^-$, $NH_4^+$, OC, and
EC). The CORR measures the correlation between the system outputs and corresponding observations, the RMSE and MAE
indicates the overall system accuracy, and the $R^2$ reflects the proportion of variability in the observations explained by the
assimilation system.





## 3 Results and discussion

### 3.1 The Dependence on Ensemble Size and Assimilation Frequency for Five Components

Ensemble size is a crucial parameter in ensemble assimilation, determining the model state's uncertainty range. A larger ensemble size more accurately represents the error distribution of state variables but requires considerable computing resources and time, especially for high-dimension systems. A smaller ensemble size can easily lead to underestimating the error covariance matrix, especially for the fine-resolution model (Kong et al., 2021). Thus, identifying an appropriate ensemble size to balance computational efficiency and accuracy is the primary step in ensemble DA. A prior study (NAQPMS-PDAF v1.0) only evaluated the correlation between ensemble size and parallel efficiency and concluded that the ratio of ensemble size to high-performance computing processors was 1:50 (Wang et al., 2022), while the impact of ensemble size on the accuracy and computational efficiency was neglected. In this study, we assessed the NAQPMS-PDAF v2.0 dependency on ensemble size through three statistical indicators (CRPS, RMSE, and CORR).

From Fig. 4a, when the ensemble size is at its minimum level of 2, the mean CRPS values of the five $PM_{2.5}$ chemical components are more significant, with $NO_3^-$ exhibiting the most considerable difference between the simulation distribution and observations (more than 4). With each increase in ensemble size, the mean CRPS values of the five chemical components progressively reduce and eventually reach convergence when the ensemble size is 10, implying that a hybrid nonlinear filter can maintain high accuracy and reliability in ensemble assimilation with an ensemble size that is smaller than the traditional minimum of 20 ensemble members, as observed in prior ensemble assimilation studies (Constantinescu et al., 2007; Miyazaki et al., 2012; Schwartz et al., 2014; Rubin et al., 2017; Kong et al., 2021; Tsikerdekis et al., 2021; Wang et al., 2022), including NAQPMS-PDAF v1.0. The mean CRPS value of EC is the lowest among the five chemical components, indicating the highest accuracy and reliability of EC ensemble DA. The performance of other components is similar. Like CRPS values, the values of RMSE and CORR decrease and increase, respectively, as the ensemble size increases, and convergence begins to occur when the ensemble size is 10 (Fig. 4b and c). Compared with other chemical components, the CORR value of $SO_4^{2-}$ is significantly lower, less than 0.8, possibly due to its estimated background field error covariance driven by the inadequate ensemble perturbations. Therefore, in the Discussion section, we deeply discuss the uncertainties of ensemble perturbations.

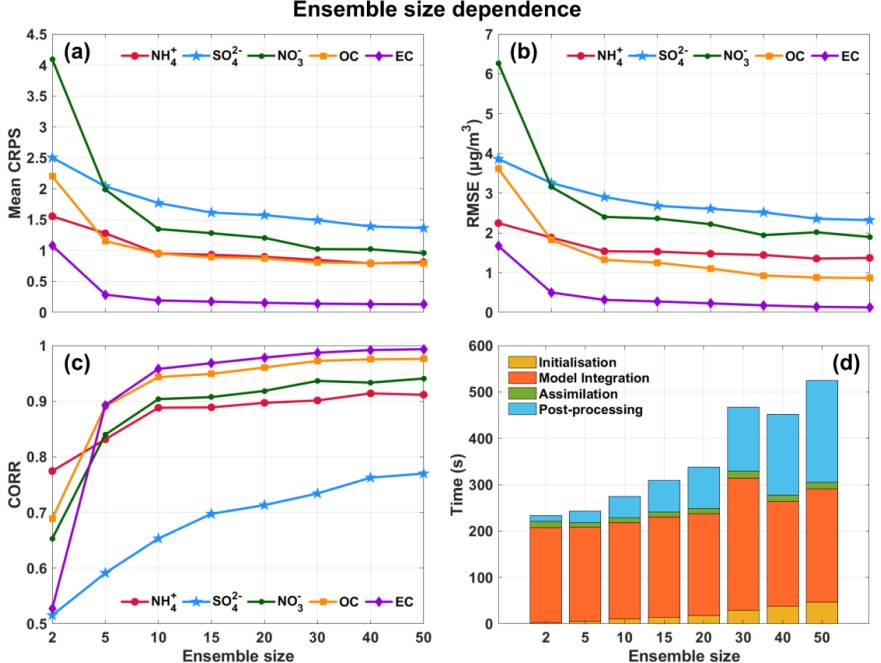

**Figure 4: Assessment of ensemble size dependency based on mean continuous ranked probability score (CRPS) (a), root mean square error (RMSE) (b), correlation coefficient (CORR) (c), and time (d).**

Figure 4d shows the time required for the four processes of ensemble assimilation under different ensemble sizes, including initialization, model integration, assimilation, and post-processing. The model integration process in NAQPMS-PDAF v2.0 takes the longest, followed by post-processing, initialization, and assimilation. The required time for initialization and post-processing increases with increasing ensemble size, while for model integration and assimilation, except for ensemble size 30, the required time is the same under different ensemble sizes. Generally, the time needed for ensemble sizes of 30-50 is considerably higher than that for smaller ones. Although convergence occurs with an ensemble size of 10, our work illustrates a similar time required between ensemble sizes 10 and 20. Consequently, we selected an ensemble size of 20 to ensure optimal performance of NAQPMS-PDAF v2.0, considering both assimilation efficiency and accuracy.

The assimilation frequency is the interval at which observational data is introduced into the DA system, directly affecting the practical assimilation data volume and computation cost. High-frequency DA with high-quality observations is crucial for improving numerical simulations and forecasts (Liu et al., 2021). Figure 5 demonstrates that the MAE values of the five chemical components analysis fields range from 0.02 to 0.12 µg/m³, RMSE values range from 0.23 to 2.61 µg/m³, and CORR values range from 0.71 to 0.98 at a 1-hour assimilation time interval, which is significantly better than the statistical indicators at lower assimilation frequencies. Even at a 2-hour assimilation frequency, the assimilation effect drops sharply compared to the 1-hour interval, especially for NO₃⁻, OC, and EC. The values of MAE and RMSE increase by 2.6-5.82 µg/m³ and 4.72-





9.57 µg/m³, respectively, and the CORR values decrease by 0.27-0.81. Gradual increasing trends in MAE and RMSE values
and a slight decreasing trend in CORR values are observed as assimilation frequency decreases from the 2-hour interval.
Therefore, the fast-updating assimilation with a 1-hour interval significantly improves the NAQPMS simulation. For the
forecasting field (Fig. S2), the low sensitivity of state variables to assimilation frequency suggests that NAQPMS-PDAF v2.0
can appropriately reduce assimilation frequency during the actual forecasting phase, lowering the demand for high temporal
resolution observations and computational resources.

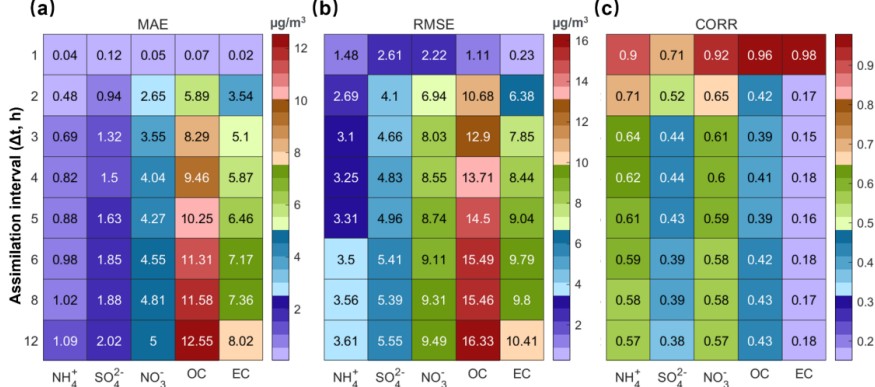


**Figure 5: Assessment of assimilation interval dependency based on mean absolute error (MAE) (a), root mean square error (RMSE)**
**(b), and correlation coefficient (CORR) (c) at the analysis step.**
**3.2 Evaluation of NAQPMS-PDAF v2.0 performance**
**3.2.1 Overall validation of DA results**
We conducted a control experiment (free-running field, FR) without any DA and a DA experiment. This section verified the
forecast filed (FOR) and analysis field (ANA) at 24 DA sites and 9 VE sites, respectively. Figure 6 shows the scatter distribution
of observations and simulations at DA sites. For FR (Fig. 6a1-a5), five chemical components have CORR values ranging from
0.32 to 0.56, and $R^2$ values do not exceed 0.3, indicating poor consistency between observations and simulations. In detail, the
simulated mass concentrations of $SO_4^{2-}$, OC, and EC are significantly overestimated, while the simulated concentrations of
$NH_4^+$ and $NO_3^-$ are underestimated. OC has the most significant error, with an RMSE value of 25.84 µg/m³ and an MAE value
of 19.41 µg/m³. Besides, the error distributions of $SO_4^{2-}$, $NO_3^-$ and $NH_4^+$ are close to a symmetric distribution with a mean
value of 0, while the error distributions of OC and EC are skewed to the left from the mean value of 0 (Fig. 7a1-a5), showing
the relatively better simulations in $SO_4^{2-}$, $NO_3^-$ and $NH_4^+$ than in OC and EC. Overall, NAQPMS cannot interpret the mass
concentrations of the five chemical components with significant errors, mainly due to the uncertainties in chemical mechanisms
(Miao et al., 2020).



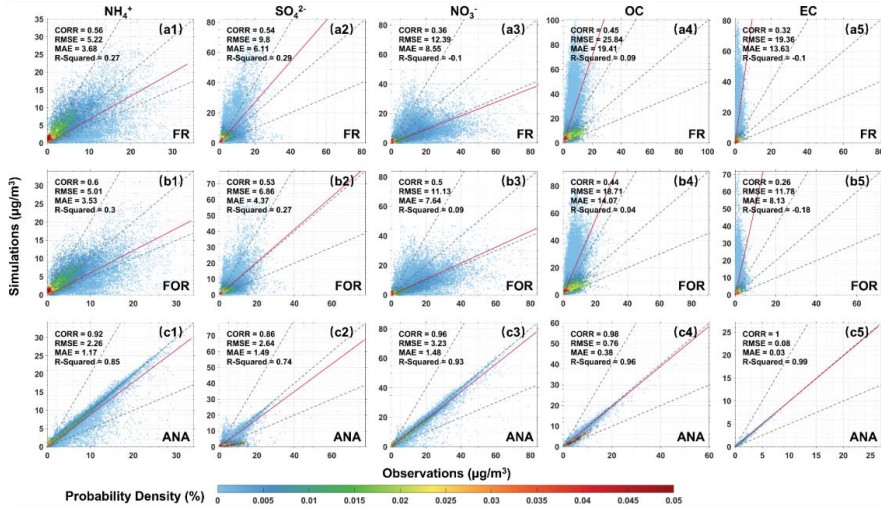

**Figure 6: Scatterplots of the DA-site simulations versus the DA-site observations with probability density for the free-running field (FR, a1-a5), forecast field (FOR, b1-b5), and analysis field (ANA, c1-c5). The dotted gray lines represent the 2:1, 1:1, and 1:2 lines, and the solid red line represents the fitting regression line.**

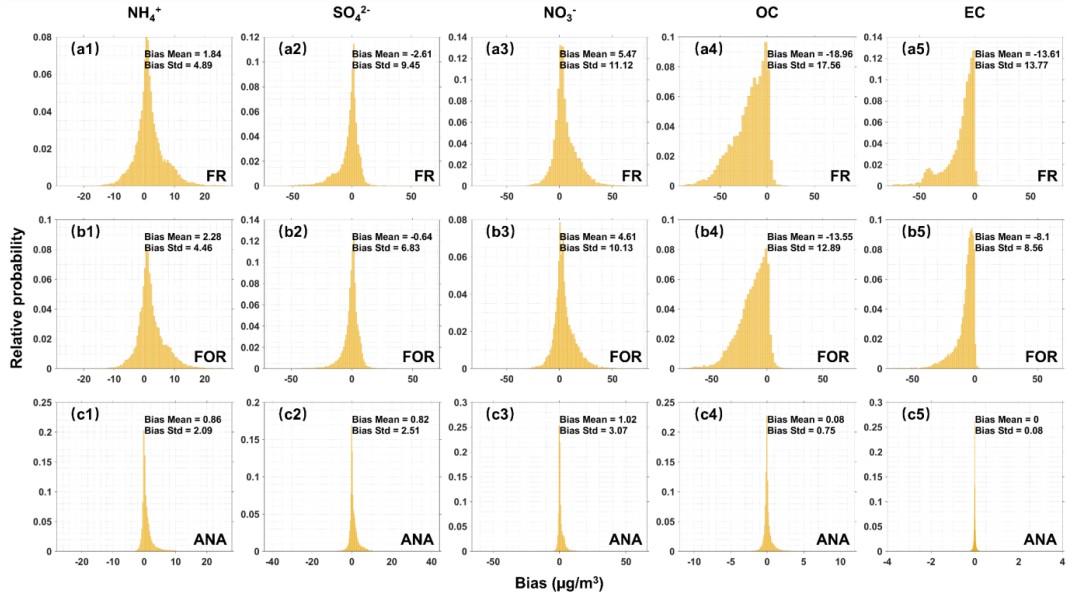

**Figure 7: Probability distributions of bias between DA-site observations and DA-site simulations for the free-running field (FR, a1-a5), forecast field (FOR, b1-b5), and analysis field (ANA, c1-c5).**

After DA, FOR shows a slight improvement with a slight increase in CORR and $R^2$ and a decrease in RMSE and MAE, especially for $NH_4^+$ and $NO_3^-$ (Fig. 6b1-b5). Although $SO_4^{2-}$, OC, and EC are significantly overestimated with a slight decrease in CORR and $R^2$, the RMSE and MAE values decrease. Besides, the error distributions of the five chemical components are concentrated at 0, and the overestimation of OC and EC has been improved compared to FR (Fig. 7b1-b5). These results indicate that DA reduces the overall FOR errors in NAQPMS due to improved forecasting ability by obtaining optimal initial





fields. However, further improvements are necessary to address the NAQPMS uncertainties in emission sources,
meteorological input, and imperfect physiochemical mechanisms. For ANA (Fig. 6c1-c5), DA significantly improves the
simulations of the five chemical components, making the ANA consistent with the observations. The CORR values are not
less than 0.86, the RMSE and MAE values do not exceed 3.23 µg/m³ and 1.49 µg/m³, respectively, and the R² values are not
less than 0.74. Specifically, the CORR values for NO₃⁻, OC, and EC are not less than 0.96, and the R² values are not less than
0.93. The error distributions of the five chemical components concentrate to 0 with the mean bias ranging from 0±0.08 µg/m³
to 1.02±3.07 µg/m³ (Fig. 7c1-c5). The results of VE sites show similar characteristics to the DA sites (Fig. S3 and S4).
Compared to FR, the overall errors of the FOR and ANA for the five chemical components decrease with a significant
improvement in ANA, showing that the CORR values of NH₄⁺ and NO₃⁻ increase by 0.15 and 0.45, respectively, the R² values
of NH₄⁺ and NO₃⁻ increase by 0.22 and 0.81, respectively, the RMSE values of OC and EC decrease by 21.77 µg/m³ and 17.79
µg/m³, respectively. Overall, the FOR and ANA errors decreased significantly. The ANA of the five chemical components at
DA sites is almost entirely consistent with the observations, indicating excellent DA performance.
**3.2.2 Assessment of temporal variation in chemical components**
The ensemble DA employs a cyclic updating process wherein the forecast and analysis steps are continuously completed at
each iteration (Evensen, 2003; Houtekamer and Zhang, 2016). In the forecast step, the ANA at the current time step serves as
the optimal initial field to advance the model integration and obtain the FOR at the next step. In the analysis step, the FOR at
the next time step provides background field information for the subsequent DA analysis to generate the ANA at the next time
step. The FOR and ANA interact with each other in the temporal dimension. Therefore, in this section, we assess the ability of
NAQPMS-PDAF v2.0 to interpret the temporal variations of the five chemical components. Figure 8 illustrates the time series
of the five chemical components at two representative sites, including a DA site in Tianjin City and a VE site in Heze City. For
the DA site (Fig. 8a), the temporal variations of NH₄⁺ and NO₃⁻ in FR and FOR exhibit better agreement with the observed
temporal variations (OBS) than those of SO₄²⁻, OC, and EC. However, NH₄⁺ and NO₃⁻ mass concentrations are significantly
lower than the high-value mass concentrations observed on February 25th. The mass concentration of SO₄²⁻ in FR is greatly
overestimated during the periods of Feb. 8th-11th, Feb. 18th-19th, and Feb. 24th-25th. The mass concentrations of OC and EC in
FR are overestimated throughout February with substantial temporal fluctuations. Although the time series of SO₄²⁻, OC, and
EC in FOR show some improvement, noticeable differences from the OBS are still apparent. After DA, the ANA time series
for the five chemical components align well with the OBS, indicating good consistency and accurate representation of temporal
characteristics, such as the NH₄NO₃ pollution captured on February 25th. Notably, the mass concentrations of SO₄²⁻, NO₃⁻, and
NH₄⁺ peaked on Feb. 8th-11th and February 25th, indicating intensified atmospheric secondary chemical reactions primarily due
to neutralization reactions of acidic pollutants capturing NH₃. The temporal variations of NH₄⁺ and NO₃⁻ are more similar
because atmospheric NO₃⁻ primarily exists as NH₄NO₃ rather than other metal nitrates, and NH₄NO₃ can form before the



complete neutralization of H₂SO₄ (Ge et al., 2017). The improvements at the VE site (Fig. 8b) are like those at the DA site,
with the ANA time series of the five chemical components showing closer agreement with the OBS, which suggests that the
localization analysis in DA effectively facilitates the propagation of observations within a specific spatial range and mitigates
the assimilation anomalies caused by spurious correlations from the distant sites (Hunt et al., 2007).

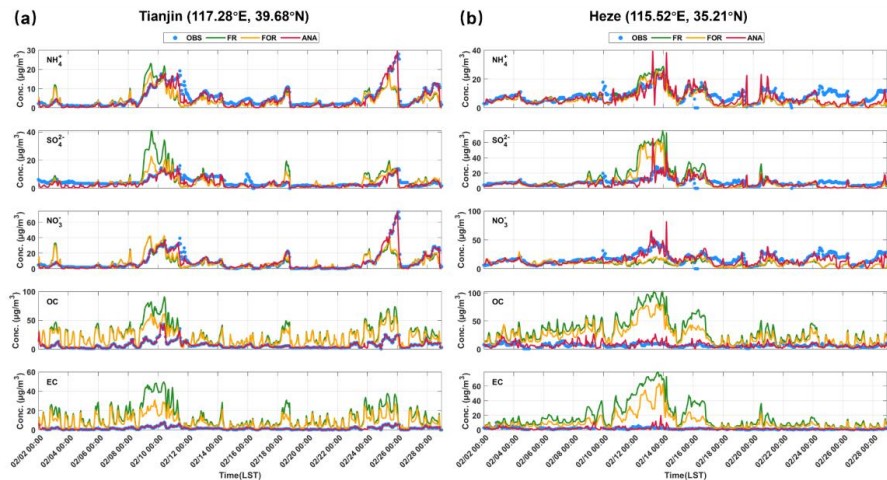


**Figure 8: Hourly variation of five PM₂.₅ chemical components in a representative DA site (a) and a representative VE site (b).**
$NH_4^+$, $SO_4^{2-}$, $NO_3^-$, OC, and EC are critical chemical components of PM₂.₅, and the sum of their mass concentrations can be
approximated as the PM₂.₅ mass concentration. We further assessed the simulation enhancement of PM₂.₅ time series based on
ground-level PM₂.₅ observations. Six representative sites were selected, including 3 DA sites (Fig. 9a1-a3) and 3 VE sites (Fig.
9b1-b3). The FR and FOR in DA and VE sites show significant overestimation and poor consistency with the OBS, mainly
due to the overestimation of OC and EC mass concentrations. Conversely, the PM₂.₅ time series in ANA closely matches that
of the OBS, accurately capturing the actual variation of PM₂.₅. In some specific instances, such as on February 26th at 00:00 in
Tianjin City and Langfang City, the peak value of ANA was lower than that of OBS, which could be attributed to the negligence
of other PM₂.₅ components (such as mineral dust and sea salt) and the inconsistency in location between ground-level PM₂.₅
observational sites and chemical components observational sites. Overall, the DA of chemical component observations
significantly enhanced the simulation of PM₂.₅ time series in NAQPMS. Compared to the CORR values of FR and FOR, the
CORR values of ANA at the six representative sites increased by 13.64%-89.58% and 17.19%-75.00%, respectively, while the
RMSE values decreased by 56.03%-83.13% and 40.74%-72.20% (Table S3).



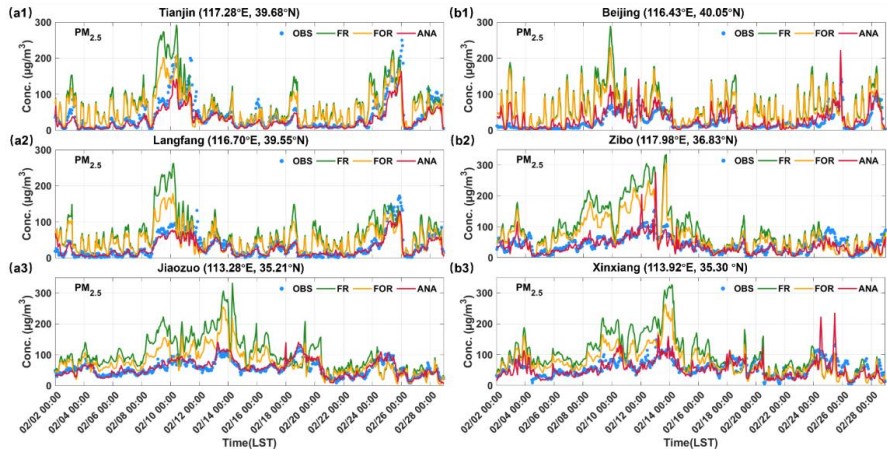


**Figure 9: Hourly variation of PM$_{2.5}$ in three representative DA sites (a1-a3) and three representative VE sites (b1-b3).**
**3.2.3 Assessment of spatial distribution in chemical components**
DA can improve the interpretation of model states in the analysis domain by using a limited number of observations. The
ability to represent spatial distribution accurately is a crucial performance for aerosol DA. Figure 10 displays the spatial
distribution of the monthly average mass concentrations for the five chemical components, including OBS, FR, FOR, ANA,
and analysis increment (INC). The spatial distributions of bias and statistical indicators for FR, FOR, and ANA are shown in
Fig. 11 and Fig. 12, respectively.

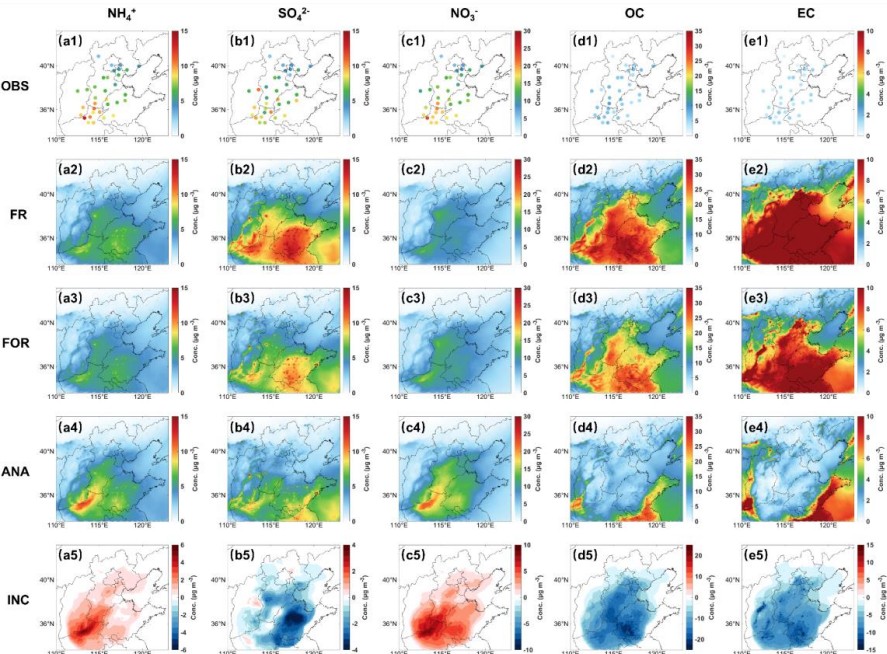


**Figure 10: Spatial concentration distribution of site observation (OBS, a1-e1), free-run field (FR, a2-e2), forecast field (FOR, a3-e3),**
**analysis field (ANA, a4-e4), and increment (INC) between ANA and FR (a5-e5) for five PM$_{2.5}$ chemical components.**



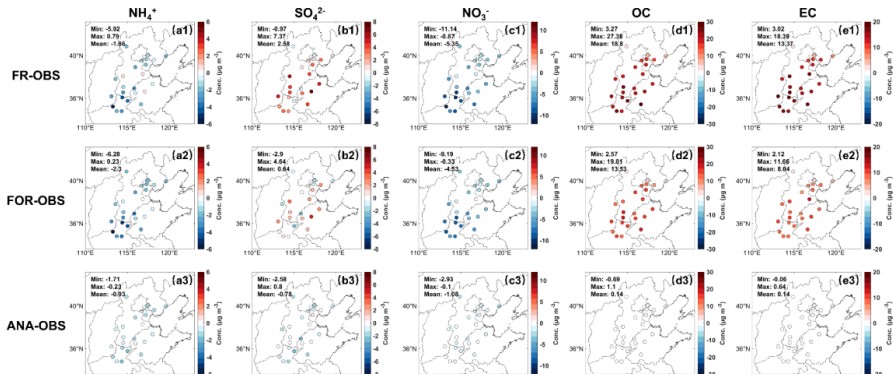


**Figure 11: Spatial distribution of DA-site bias for five PM$_{2.5}$ chemical components from observation (OBS) for the free-running field**


**(FR, a1-e1), forecast field (FOR, a2-e2) and analysis field (ANA, a3-e3).**


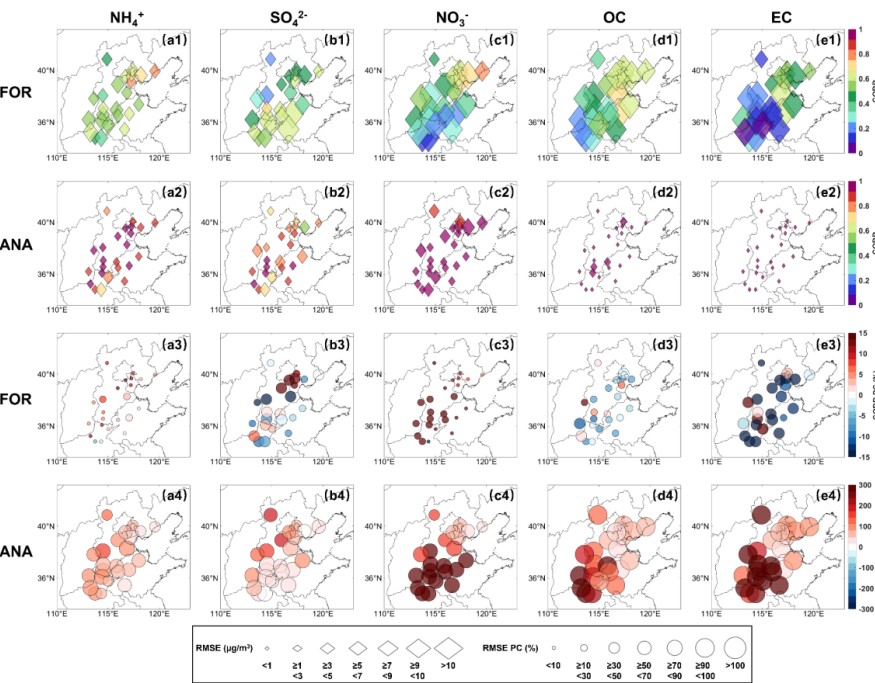


**Figure 12: Spatial distribution of DA-site statistical indictors for five PM$_{2.5}$ chemical components. (a1-e1) represents the values of**


**RMSE and CORR for the forecast field (FOR), (a2-e2) same as (a1-e1) but for analysis field (ANA), (a3-e3) represents the**


**improvement of RMSE and CORR for the forecast field (FOR), (a4-e4) same as (a3-e3) but for analysis field (ANA). The size**


**represents the value of RMSE in (a1-e2) and the improvement percentage compared to non-assimilation in (a3-e4), respectively.**


The spatial characteristics of NH$_4^+$ and NO$_3^-$ are similar. Compared to the OBS (Fig. 10a1 and c1), the FR (Fig. 10a2 and c2)


and FOR (Fig. 10a3 and c3) have failed to capture the high-value mass concentrations in the border area between Hebei


province, Shanxi province, Henan province, and Shandong province, especially in the northern region of Henan province. The


primary reason is the uncertainties in emission inventories in winter heating periods, which results in insufficient emission


statistics of gaseous precursors NOx and NH$_3$ (Aleksankina et al., 2018). After DA, this situation is significantly improved






with the ANA (Fig. 10a4 and c4). The INCs in the Beijing-Tianjin-Hebei region, Shanxi province, Henan province, and
Shandong province are positive (Fig. 10a5 and c5), indicating varying degrees of improvement in correcting the
underestimation of mass concentrations. Specifically, for $NH_4^+$ and $NO_3^-$ at DA sites, the biases between the OBS and ANA
are significantly reduced compared to the biases between the OBS and FR (Fig. 11), with the mean absolute bias decreasing
by 0.93 μg/m$^3$ and 4.27 μg/m$^3$, respectively. Moreover, the overall biases at VE sites also decrease (Fig. S5). As for the spatial
statistical indicators of $NH_4^+$ (Fig. 12a1 and a2), the CORR values in FOR and ANA range from 0.39 to 0.79 and 0.70 to 0.97,
respectively, and the RMSE values range from 3.16 μg/m$^3$ to 7.65 μg/m$^3$ and 1.20 μg/m$^3$ to 3.49 μg/m$^3$, respectively. As for
the spatial statistical indicators of $NO_3^-$ (Fig. 12c1 and c2), the CORR values in FOR and ANA range from 0.09 to 0.76 and
0.89 to 0.99, respectively, and the RMSE values range from 4.88 μg/m$^3$ to 15.69 μg/m$^3$ and 1.34 μg/m$^3$ to 5.39 μg/m$^3$,
respectively. For the FOR, the improvement in accuracy for $NO_3^-$ is more significant than that for $NH_4^+$, with the CORR values
of most DA sites increasing by more than 10% and the RMSE of most DA sites decreasing by not less than 10% (Fig. 12a3
and c3). For the ANA, $NH_4^+$, and $NO_3^-$ exhibit significant improvements in CORR and RMSE, as most DA sites show over
150% in CORR and over 50% in RMSE (Fig. 12a4 and c4). The improvements can also be found for $NH_4^+$ and $NO_3^-$ at VE
sites (Fig. S6). The spatial consistency of $NH_4^+$ and $NO_3^-$ indicates that $NH_4NO_3$ is the primary aerosol chemical component,
highlighting the necessity of coordinated control of precursor NOx and $NH_3$.

Unlike $NH_4^+$ and $NO_3^-$, compared to the OBS (Fig. 10b1), the mass concentrations of $SO_4^{2-}$ in the FR and FOR (Fig. 10b2 and
b3) are significantly overestimated, especially in Shandong province. In contrast, the ANA has dramatically improved (Fig.
10b4), with most areas showing negative INCs (Fig. 10b5). The mean absolute biases in DA and VE sites have decreased by
1.80 μg/m$^3$ and 2.68 μg/m$^3$, respectively (Fig. 11 and Fig. S5). Specifically, after DA, the CORR values of the FOR and ANA
range from 0.22 to 0.71 and 0.58-0.97, and the RMSE values range from 3.42 μg/m$^3$ to 11.07 μg/m$^3$ and 1.20 μg/m$^3$ to 4.30
μg/m$^3$, respectively (Fig. 12b1 and b2). The CORR and RMSE values in FOR have significantly improved (Fig. 12b3) at DA
sites around Beijing. While the CORR values in ANA have increased by more than 13%, with most DA sites showing an
increase of over 50%, and RMSE values have decreased by no less than 30%, with most DA sites showing a decrease of over
70% (Fig. 12b4). Besides, half of the VE sites show significant improvement in the CORR and RMSE in the FOR and ANA,
mainly due to their proximity to more DA sites (Fig. S6). The OBS and ANA indicate a considerable control in $SO_4^{2-}$ pollution
during the winter heating period due to the emission reduction of gaseous precursors (Zhai et al., 2019; Yan et al., 2021).

The spatial distributions of OC and EC exhibit similarities (Fig. 10d1 and e1), consistent with the finding of a strong correlation
between OC and EC in winter (Cao et al., 2007). Since the low temperature and weakened photochemical reactions in winter
reduced secondary OC (SOC) generation, and primary OC (POC) and EC mainly originate from direct anthropogenic
emissions, such as combustion (Guo, 2016). Compared to the OBS, the mass concentrations in FR (Fig. 10d2-d3) and FOR



(Fig. 10e2-e3) are significantly overestimated over a wide range. Similar overestimations have also been reported in the global
reanalysis datasets of CAMS and MERRA-2, likely attributed to the hygroscopic growth scheme of carbonaceous aerosols in
the models, poorly constrained semi-volatile species escaping from primary organic aerosols (Soni et al., 2021), and aging
mechanisms in the models (Huang et al., 2013). After DA, the spatial distribution of the ANA aligns entirely with that of the
OBS (Fig. 10d4 and e4), with the improvements in all overestimations (Fig. 10d5 and e5) and the average biases of OC and
EC at DA sites both significantly decreasing to 0.14 µg/m³ (Fig. 11d3 and e3). The VE sites show similar results to the DA
sites, with the average biases of less than 2 µg/m³ (Fig. S5d3 and e3). Specifically, for OC (Fig. 12d1 and d2), the CORR
values in FOR and ANA are 0.18-0.71 and 0.92-1.00, respectively, with RMSE values of 7.91 µg/m³-26.27 µg/m³ and 0.16
µg/m³-1.45 µg/m³, respectively. For EC (Fig. 12e1 and e2), the CORR values in FOR and ANA are 0.01-0.66 and 0.97-1.00,
respectively, with RMSE values of 5.33 µg/m³-16.91 µg/m³ and 0.01 µg/m³-0.26 µg/m³, respectively. Although significant
improvements are not observed in FOR at some specific DA sites, the RMSE values at all DA sites decrease by 10%-50% (Fig.
12d3 and e3). The CORR values of OC and EC in ANA increase by more than 30%, with most DA sites exceeding 200%, and
the RMSE values decrease by more than 90% (Fig. 12d4 and e4). At VE sites (Fig. S6), significant improvements in the CORR
are not observed, but the RMSE values in the FOR and ANA decrease, which indicates that DA has limited benefits for whole
areas but can effectively reduce biases of whole areas.
**3.3 Compared to NAQPMS-PDAF v1.0 and global reanalysis dataset**
To comprehensively evaluate the competitiveness and superiority of NAQPMS-PDAF v2.0 in generating the reanalysis
datasets of the PM$_{2.5}$ chemical compositions, we assimilated the mass concentrations of the five PM$_{2.5}$ chemical components
from all sites (sum of DA sites and VE sites) in February 2022 to generate a reanalysis dataset. We compared our reanalysis
dataset with the global reanalysis (RA) datasets (CAMSRA and MERRA-2) and NAQPMS-PDAF v1.0 output. Figure 13
illustrates the spatial distribution of the monthly average mass concentrations for the five chemical components. Compared to
the OBS (Fig. 13a1 and c1), CAMSRA underestimates the NH$_4^+$ and NO$_3^-$ concentrations and fails to capture the high-value
concentration in northern Henan Province (Fig. 13a2 and c2). Meanwhile, MERRA-2 overestimates the concentrations of
SO$_4^{2-}$, OC, and EC (Fig. 13b2, d2 and e2), particularly SO$_4^{2-}$, exhibiting a large region with inaccurately high concentrations.
Besides, CAMSRA (approximately 80*80 km$^2$) and MERRA-2 (55*70 km$^2$) have significantly lower spatial resolutions
compared to NAQPMS-PDAF v2.0 (5*5 km$^2$). Therefore, NAQPMS-PDAF v2.0 provides a more detailed description of the
pollution characteristics of chemical components in Northern China and surrounding areas compared to RA.





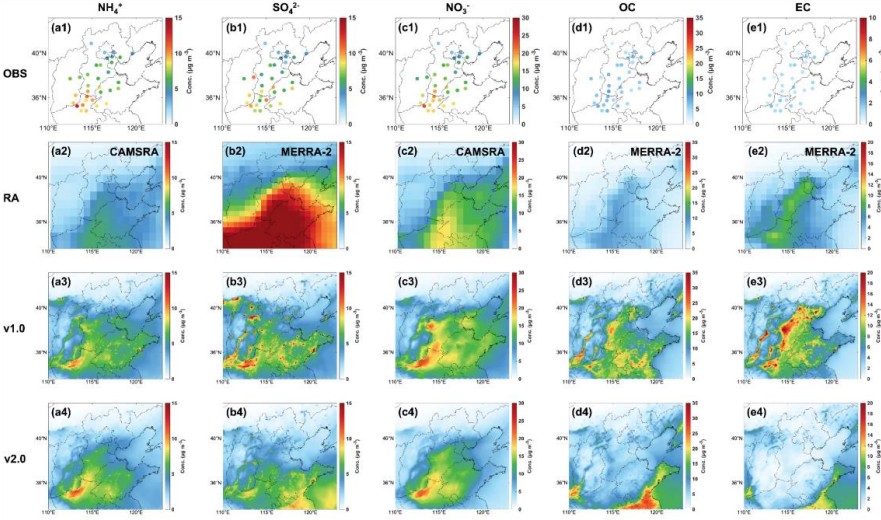


**Figure 13: Spatial distribution of the monthly averaged concentration of five PM$_{2.5}$ chemical components for observations (OBS, a1-**
**e1), global reanalysis data (RA, a2-e2), NAQPMS-PDAF v1.0 analysis data (a3-e3) and NAQPMS-PDAF v2.0 analysis data (a4-e4).**
Although NAQPMS-PDAF v1.0 demonstrates a superior spatial representation of the five chemical components when
compared to RA, it fails to capture the high-value concentrations of NH$_4^+$ in the northwest of Henan Province and correct the
high-value concentrations of NH$_4^+$ in the central and western areas of Hebei Province (Fig. 13a3). Moreover, the scattered
high-value concentrations of SO$_4^{2-}$ in the North China Plain do not align with the spatial characteristics of the OBS (Fig. 13b3).
Notably, NAQPMS-PDAF v1.0 exhibits poor performance in interpreting OC and EC with significant overestimations in a
wide range (Fig. 13d3 and e3), which indicates that NAQPMS-PDAF v1.0 is weaker than NAQPMS-PDAF v2.0 in terms of
DA performance on chemical components, primarily due to insufficient propagation of observations. In NAQPMS-PDAF v2.0,
the LKNETF algorithm with an adaptive forgetting factor is more suitable for the nonlinear and non-Gaussian situations
compared to EnKFs in NAQPMS-PDAF v1.0, and the ensemble perturbation with non-Gaussian distribution can better
represent the reasonable error distribution of model states.

Table 3 presents a quantitative comparison of three reanalysis datasets. Compared to the CORR of NAQPMS-PDAF v2.0
(0.86-0.99), the CORR of RA for the five chemical components is significantly lower (0.42-0.55). Moreover, NAQPMS-PDAF
v1.0 exhibits significantly poorer consistency in SO$_4^{2-}$, OC, and EC, with CORR values ranging from 0.35 to 0.57. NAQPMS-
PDAF v2.0 has lower overall RMSE values (0.14 µg/m$^3$-3.18 µg/m$^3$) compared to RA and NAQPMS-PDAF v1.0, with RMSE
values ranging from 4.51 µg/m$^3$ to 12.27 µg/m$^3$ and 2.46 µg/m$^3$ to 15.50 µg/m$^3$, respectively. The characteristics of the R$^2$ are
like those of the CORR and RMSE. For NH$_4^+$ and NO$_3^-$, NAQPMS-PDAF v2.0 (0.85 and 0.93) and v1.0 (0.80 and 0.96) are
much higher than RA (0.09 and 0.13). Notably, for SO$_4^{2-}$, OC, and EC, NAQPMS-PDAF v2.0 (0.74-0.98) is significantly
higher than v1.0 (-0.16-0.25) and RA (-0.15-0.25). Overall, NAQPMS-PDAF v2.0 more accurately and consistently interprets



the five chemical components, particularly for $NH_4^+$, $SO_4^{2-}$, OC, and EC. The reasons are summarized as follows. (1) The DA frequency of CAMSRA is 12 hours, which is lower than the hourly DA frequency in NAQPMS-PDAF v2.0. (2) CAMSRA only assimilates satellite retrievals (Inness et al., 2019), and MERRA-2 only assimilates aerosol optical depth (AOD) from both ground-based and space-based remote sensing platforms (Randles et al., 2017). The aerosol optical information analysis increment cannot be allocated to each chemical component accurately and reasonably due to the lack of a deterministic relationship between aerosol optical information and $PM_{2.5}$ chemical components. (3) NAQPMS-PDAF v1.0 has evident DA shortcomings for chemical components due to the limited DA algorithm under the assumption of linear model or system, inappropriate ensemble perturbation under the assumption of Gaussian distribution, and inadequate observational modules. (4) The state variable structure in NAQPMS-PDAF v1.0 lacks the capacity to effectively mitigate the impact of spurious correlations between chemical component variables, even when using analytical localization.

**Table 3: Statistical indicators (CORR, RMSE, $R^2$) of five $PM_{2.5}$ chemical components for global reanalysis data (RA), NAQPMS-PDAF v1.0 analysis data and NAQPMS-PDAF v2.0 analysis data.**

| Components | CORR | | | RMSE ($\mu g/m^3$) | | | $R^2$ | | |
|---|---|---|---|---|---|---|---|---|---|
| | RA | v1.0 | v2.0 | RA | v1.0 | v2.0 | RA | v1.0 | v2.0 |
| $NH_4^+$ | 0.49 | 0.90 | 0.92 | 5.59 | 2.53 | 2.22 | 0.09 | 0.80 | 0.85 |
| $SO_4^{2-}$ | 0.55 | 0.57 | 0.86 | 12.27 | 5.45 | 2.61 | 0.25 | 0.25 | 0.74 |
| $NO_3^-$ | 0.54 | 0.98 | 0.96 | 10.27 | 2.46 | 3.18 | 0.13 | 0.96 | 0.93 |
| OC | 0.50 | 0.42 | 0.97 | 4.51 | 12.92 | 0.93 | 0.15 | -0.09 | 0.93 |
| EC | 0.42 | 0.35 | 0.99 | 7.59 | 15.50 | 0.14 | -0.15 | -0.16 | 0.98 |

### 3.4 The uncertainty in NAQPMS-PDAF v2.0

In ensemble DA, the ensemble members represent possible values of the model states, and the ensemble sampling can determine the uncertainties of the model states. Therefore, the ensemble generation directly affects the propagation of observations and subsequently impacts the final DA performance. Previous studies generated ensemble members based on the uncertainties of emission species and the Gaussian-distribution assumption to satisfy the requirements of EnKFs algorithms (Kong et al., 2021; Wang et al., 2022). However, the true error probability distribution of emission species is not an ideal Gaussian distribution, and the assumption will introduce errors. In this study, we coupled the hybrid nonlinear DA algorithm (LKNETF) with NAQPMS to handle the nonlinear and non-Gaussian situations, which combines the stability of LETKF with the nonlinearity of LNETF. Therefore, we evaluate the performance of ensemble members with different uncertainties and error probability distributions in NAQPMS-PDAF v2.0 through two groups of sensitivity experiments.

The first group of experiments (T1-T5) involves controlling the $SO_2$ uncertainty as a fixed value of 200% and transforming the distribution of the perturbation coefficient matrix. The second group of experiments (M1-M5) focuses on assessing the influence of $SO_2$ uncertainty on $NH_4^+$ and $SO_4^{2-}$ DA based on a fixed non-Gaussian distribution (m3=1, m4=6). Figure 14



shows the statistical indicators of the five chemical components under different error probability distributions, including a
Gaussian distribution (T1) and four non-Gaussian distributions (T2-T5). The mean CRPS and RMSE in T2 and T4 are lower
than those in T1, T3, and T5, and the CORR values in T2 and T4 are higher than those in T1, T3, and T5, indicating that the
DA performance of non-Gaussian-distribution assumption is superior to that of Gaussian-distribution assumption. Moreover,
positively skewed non-Gaussian distribution performs better than negatively skewed distribution. Except for $SO_4^{2-}$, the
performance in T2 outweighs that in T4 for other chemical components, implying that higher kurtosis harms the chemical
components DA.

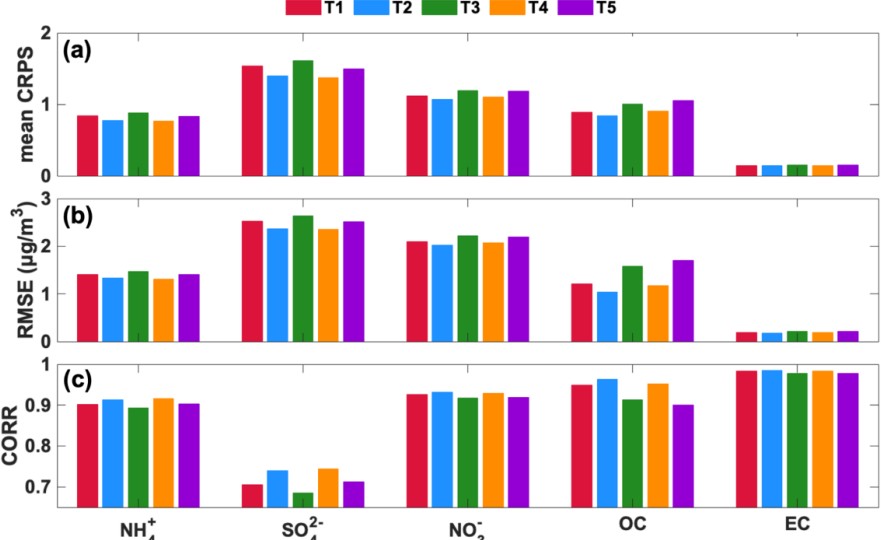


**Figure 14: Statistical indicators (mean CRPS (a), RMSE (b), and CORR (c)) of five $PM_{2.5}$ chemical components for five perturb**
**experiments based on distribution.**
$SO_2$ is a crucial precursor of $NH_4^+$ and $SO_4^{2-}$, and perturbing $SO_2$ affects the forecast and simulation of $NH_4^+$ and $SO_4^{2-}$. Table
4 presents statistical indicators of $NH_4^+$ and $SO_4^{2-}$ analysis fields based on ensemble perturbations with different $SO_2$
uncertainties (12%-300%). Increasing the uncertainty of $SO_2$ from 12% to 200% leads to a decrease in the mean CRPS in the
$SO_4^{2-}$ analysis field from 2.67 to 1.40, an increase in the CORR from 0.51 to 0.74, and a reduction in the RMSE from 4.10
$\mu g/m^3$ to 2.37 $\mu g/m^3$. Similarly, the mean CRPS in the $NH_4^+$ analysis field decreases from 0.98 to 0.77, the CORR increases
from 0.88 to 0.91, and the RMSE decreases from 1.55 $\mu g/m^3$ to 1.33 $\mu g/m^3$. It indicates that increasing the uncertainty of $SO_2$
improves the DA performance on $NH_4^+$ and $SO_4^{2-}$ because the higher $SO_2$ uncertainty makes $SO_2$ perturbed sufficiently, and
the estimated error probability distribution is closer to the real distribution, resulting in a sufficient spread of observations.
However, when the uncertainty of $SO_2$ reaches 300%, the statistical indicators do not significantly improve and even worsen
because excessively high $SO_2$ uncertainty causes the estimated error probability distribution to deviate from the true
distribution. Thus, selecting appropriate uncertainties for emission species is crucial in aerosol chemical component DA.






To summarize, the non-Gaussian-distribution assumption outperforms the Gaussian-distribution assumption in NAQPMS-
PDAF v2.0. Positive skewness performs better than negative skewness, and excessively high kurtosis should be avoided.
Additionally, appropriately increasing the uncertainty of $SO_2$ enhances the DA performance of $NH_4^+$ and $SO_4^{2-}$. Future studies
should conduct more sensitivity experiments on emission species perturbation to determine the suitable schemes for different
aerosol chemical components.
**Table 4: Statistical indicators (mean CRPS (a), RMSE (b), and CORR (c)) of five PM$_{2.5}$ chemical components for five perturb**
**experiments based on SO$_2$ emission uncertainty.**

| Experiment | $SO_4^{2-}$ | | | $NH_4^+$ | | |
|---|---|---|---|---|---|---|
| | CRPS | CORR | RMSE | CRPS | CORR | RMSE |
| M1 | 2.67 | 0.51 | 4.10 | 0.98 | 0.88 | 1.55 |
| M2 | 2.07 | 0.59 | 3.24 | 0.92 | 0.89 | 1.48 |
| M3 | 1.61 | 0.69 | 2.63 | 0.83 | 0.91 | 1.39 |
| M4 | 1.40 | 0.74 | 2.37 | 0.77 | 0.91 | 1.33 |
| M5 | 1.41 | 0.74 | 2.39 | 0.78 | 0.91 | 1.33 |

**4 Conclusions**
In this paper, we online coupled NAQPMS with PDAF-OMI to develop a novel hybrid nonlinear DA system (NAQPMS-
PDAF v2.0) with level-2 parallelization based on a hybrid Kalman-Nonlinear Ensemble Transform Filter (LKNETF) for the
first time. Compared to NAQPMS-PDAF v1.0, NAQPMS-PDAF v2.0 with OMI can be applied with multiple component
types and nonlinear/non-Gaussian situations in chemical analysis to effectively interpret five PM$_{2.5}$ chemical components
($NH_4^+$, $SO_4^{2-}$, $NO_3^-$, OC and EC), which is not achieved in previous studies. The background error covariance was calculated
by ensemble perturbation based on adaptive uncertainties and non-Gaussian-distribution assumption of emission species. The
DA experiments were conducted based on 33 observational sites in Northern China and surrounding areas.

NAQPMS-PDAF v2.0 with LKNETF can maintain high accuracy and reliability in ensemble DA with an ensemble size of 10,
smaller than the traditional minimum of 20 ensemble members, as observed in prior ensemble assimilation studies. The FR
(free-run fields without DA) have a poor consistency with the observations, with the CORR values ranging from 0.32-0.56
and the $R^2$ values less than 0.3, showing that $SO_4^{2-}$, OC and EC are significantly overestimated, while $NH_4^+$ and $NO_3^-$ are
underestimated. A significant improvement was observed in the ANA (analysis fields) of the DA sites. The CORR values are
not less than 0.86, the RMSE and MAE values do not exceed 3.23 $\mu g/m^3$ and 1.49 $\mu g/m^3$, respectively, and $R^2$ is not less than
0.74. Specifically, the CORR values for $NO_3^-$, OC, and EC are not less than 0.96, and $R^2$ is not less than 0.93. The error
distributions of the five chemical components concentrate to 0 with the mean bias ranging from 0±0.08 $\mu g/m^3$ to 1.02±3.07
$\mu g/m^3$. These improvements are also found in the ANA at VE sites, indicating an excellent DA performance of NAQPMS-

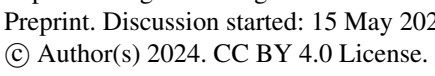



PDAF v2.0.

The ability of NAQPMS-PDAF v2.0 to interpret the spatiotemporal characteristics of the five chemical components was
examined. For temporal variations, compared to the FR and FOR (forecast fields), the ANA closely aligned with the OBS
(observations) and accurately captured the peak concentrations of $SO_4^{2-}$, $NO_3^-$, and $NH_4^+$ on specific periods (such as February
25$^{th}$), indicating good consistency and accurate characterization. Specifically, the CORR of the ANA at the six representative
sites increased by 13.64%-89.58% and 17.19%-75.00%, respectively, while the RMSE decreased by 56.03%-83.13% and
40.74%-72.20%. For spatial distributions, after DA, both $NH_4^+$ and $NO_3^-$ with positive analysis increments exhibit significant
improvements in CORR and RMSE, as most DA sites show improvements of over 150% in CORR and over 50% in RMSE.
$SO_4^{2-}$, OC, and EC with negative analysis increments were also improved. Especially for OC and EC, the improvements of
CORR and RMSE at most DA sites were over 200% and over 90%, respectively. The improvements at VE sites were also
identified. Consequently, DA successfully aligned the spatiotemporal characteristics of the ANA with OBS and significantly
reduced the biases of five chemical components.

Compared to the global reanalysis datasets (CORR: 0.42-0.55, RMSE: 4.51-12.27 µg/m$^3$) and NAQPMS-PDAF v1.0 (CORR:
0.35-0.98, RMSE: 2.46-15.50 µg/m$^3$), the NAQPMS-PDAF v2.0 (CORR: 0.86-0.99, RMSE: 0.14-3.18 µg/m$^3$) has significant
superiority in generating the reanalysis datasets of the PM$_{2.5}$ chemical compositions with high spatiotemporal resolution.
Besides, NAQPMS-PDAF v1.0 cannot capture the high-value concentrations and exhibits poor performance when interpreting
$SO_4^{2-}$, OC, and EC with CORR values ranging from 0.35 to 0.57. In contrast, NAQPMS-PDAF v2.0 interprets the five chemical
components more accurately and consistently.

Finally, the uncertainties of NAQPMS-PDAF v2.0 are examined by identifying the influence of ensemble generation on
ensemble DA performance. The non-Gaussian-distribution assumption outperforms the Gaussian-distribution assumption in
NAQPMS-PDAF v2.0. Positive skewness performs better than negative skewness, and excessively high kurtosis should be
avoided. Additionally, appropriately increasing the uncertainty of $SO_2$ enhances the DA performance of $NH_4^+$ and $SO_4^{2-}$. Future
studies should conduct more sensitivity experiments on emission species perturbation to determine the suitable schemes for
different aerosol chemical components.

The novel hybrid nonlinear DA system (NAQPMS-PDAF v2.0) can be effectively applied in the interpretation of chemical
components and outperform in generating the reanalysis dataset of the five PM$_{2.5}$ chemical components with high accuracy
and high consistency, thus providing the sufficient channel to investigate the spatiotemporal characteristics, identify the
regional transport and prevent and control aerosol composition pollution. In future work, we plan to research the vertical DA



of chemical components, introduce more vertical information from more observational platforms, and verify the simultaneous
DA performance of surface and vertical mass concentrations.

**Code and data availability**

The source codes in our work are available online via Zenodo (https://doi.org/10.5281/zenodo.10886914).

**Author contributions**

HL developed the data assimilation system, performed numerical experiments, carried out the analysis and wrote the original
manuscript. TY provided scientific guidance, designed the paper strcutre and wrote this paper. LN developed PDAF and
provided help for the model code. DWZ, DZ, and GT provided $PM_{2.5}$ chemical component data. HW provided help for the
model code. YS, PF, HS, ZW did overall supervision. All authors reviewed and revised this paper.

**Competing interests**

The contact author has declared that neither they nor their co-authors have any competing interests.

**Acknowledgements**

This work was supported by the National Key Research and Development Program for Young Scientists of China (No.
2022YFC3704000), the National Natural Science Foundation of China (No. 42275122) and the National Key Scientific and
Technological Infrastructure project "Earth System Science Numerical Simulator Facility" (EarthLab). Ting Yang would like
to express gratitude towards the Program of the Youth Innovation Promotion Association (CAS). We thank the Big Data Cloud
Service Infrastructure Platform (BDCSIP) for providing computing resources.



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
