# Peer review of "NAQPMS-PDAF v2.0: A Novel Hybrid Nonlinear Data Assimilation System for Improved Simulation of PM2.5 Chemical Components"

_Geoscientific Model Development, 2024_

## Author Comment (AC1)

**Authors' responses to Referees' comments**

**Journal:** Geoscientific Model Development

**Manuscript Number:** gmd-2024-78

**Title:** NAQPMS-PDAF v2.0: A Novel Hybrid Nonlinear Data Assimilation System for Improved Simulation of PM$_{2.5}$ Chemical Components

**Authors:** Hongyi Li, Ting Yang, et al.

Note:

Comment (12-point black italicized font).

Reply (indented, 12-point blue normal font).

"Revised text as it appears in the text (in quotes, 12-point blue italicized font)".
* * *
**Anonymous Referee #1**

*1 General comments:*

*This manuscript integrates an ensemble Kalman filter-based non-linear data assimilation method with an atmospheric chemistry transport model (CTM). The primary advancement is the coupling of the hybrid Kalman-Nonlinear Ensemble Transform Filter (KNETF) with an adaptive forgetting factor to the CTM model. The method was tested using a real-world dataset, with experiments varying ensemble sizes and evaluated against multiple metrics. **The presentation quality is good, though some minor issues need to be addressed**.*

**Authors' response:**

We thank the reviewer for the positive assessment and constructive suggestions of our manuscript.

*2 Detailed Comments:*

*1) Line 15: Replace "difficulty" with "challenge".*

**Authors' response:**

We thank the reviewer for the suggestion, and we agree that "challenge" is more accurate. The revised text is shown below.

Abstract, Line 15: "*However, accurately describing spatiotemporal variations of PM$_{2.5}$*

*chemical components remains a challenge.*"

*2) Line 147: The term "level-2" is not adequately introduced. Consider moving the reference from Lines 152-153 to the beginning of the paragraph for clarity.*

**Authors' response:**

We thank the reviewer for the suggestion and agree that the reference from Lines 152-153 should be moved to the beginning of the paragraph for clarity. Since the term "level-2" is detailed in our previous work (Wang et al. 2022), this manuscript dose not adequately introduce.

For NAQPMS-PDAF v2.0, we designed the level-2 parallel computational framework as in NAQPMS-PDAF v1.0 by using the Message Passing Interface (MPI) standard to ensure high computational efficiency in ensemble data assimilation. The level-2 parallel computational framework can simultaneously perform the parallel computation within the atmospheric chemistry transport model NAQPMS and the parallel computation of the ensemble tasks. For example, running 20 ensembles means executing 20 model tasks, each of which requires integral computation in a large model grid. With enough computational resources, 20 model tasks can be executed simultaneously at different computational nodes, and each model task can perform parallel computation in the large model grid with multiple processors. For practice, NAQPMS-PDAF v2.0 initializes the main communicator (MPI_COMM_WORLD) into three sub-communicators (Fig. R1), namely COMM_model, which is responsible for model integration, COMM_filter, which is responsible for the filter analysis, and COMM_couple, which is responsible for the information transfer between the model and the filter. Taking the example of calling 12 processors, assuming that the ensemble size is 3, NAQPMS-PDAF v2.0 will initialize 3 COMM_models for 3 model tasks, and each model task can be assigned 4 processors. Then, each model grid can be cut into 4 sub-grids for parallel computation. COMM_couple combines the communicators for different model tasks. COMM_filter occupies the same number of processors as the first model task (COMM_model 1) with 4 processors.

[Figure]

**Figure R1: Example of a typical configuration of the communicators using a parallelized model (quoted from https://pdaf.awi.de/trac/wiki)**

According to the reviewer's suggestion, we moved the reference to the beginning of the paragraph for clarity and added the description to the term "level-2". The revised text is shown below.

Section 2.3.1, Line 146-153: "*NAQPMS-PDAF v2.0 implements an online coupling between NAQPMS and PDAF v2.1 with OMI, utilizing a level-2 parallel computational framework. The description of level-2 parallel implementation was detailed in our previous work (Wang et al., 2022). The online coupling ensures the continuous operation of model forecasts and assimilation analysis at each time step, achieved by directly integrating PDAF routines into the prototype code of NAQPMS (the right portion of Fig. 1, the blue represents NAQPMS main routines, while the yellow represents PDAF main routines). The level-2 parallel computational framework, which utilizes the Message Passing Interface standard (MPI), facilitates concurrent processing and data exchange among multiple ensemble members and parallel computation among model state matrixes within each ensemble member, enhancing the efficiency of ensemble analysis and numerical model computations. For instance, the operation of twenty ensemble members necessitates the execution of twenty model tasks, each of which performs integral calculations on a large model grid. Twenty model tasks can be executed simultaneously at twenty computational nodes with sufficient computational resources. Each model task can then perform parallel computation with multiple processors by splitting the large model grid into multiple sub-grids.*"

**Reference**

Wang, H., Yang, T., Wang, Z., Li, J., Chai, W., Tang, G., Kong, L., and Chen, X.: An aerosol vertical data assimilation system (NAQPMS-PDAF v1.0): development and application, Geosci. Model Dev., 15, 3555-3585, https://doi.org/10.5194/gmd-15-3555-2022, 2022.

*3) Line 215: Properly cite online resources instead of directly inserting hyperlinks.*

**Authors' response:**

We thank the reviewer for the reminder and apologize for the inappropriate citation. The revised text is shown below.

==Section 2.3.3, Line 214-215:== "*...which are subsequently transformed into non-Gaussian distribution matrixes through non-Gaussian process generation v1.2 (Cheynet, 2024).*"

**Reference**

Cheynet, E.: Non-Gaussian process generation, https://github.com/ECheynet/Gaussian_to_nonGaussian/releases/tag/v1.2, GitHub. Retrieved July 7, 2024.

*4) Figure 3: The coloring of Domain 3 is difficult to distinguish.*

**Authors' response:**

We thank the reviewer for the suggestion, and we revised the Fig. 3 in the original manuscript by changing the line color of domain 3. The revised version is show as below.

[Figure]

*Figure 3: The model domains in WRF simulation (a) and the location of observations*

*(b). Domain 3 in (a) is the target area of this study. Twenty-four red sites in (b) represent the sites for data assimilation, and nine green sites in (b) represent the sites for spatial independent validation. The topographic dataset is from the ETOPO1 1 arc-minute Global Relief Model, taken from the National Geophysical Data Center (Amante and Eakins, 2009).*

**Reference**

Amante, C. and Eakins, B. W.: ETOPO1 arc-minute global relief model: procedures, data sources and analysis, 2009.

*5) Line 395: Remove the word "deeply".*

**Authors' response:**

We thank the reviewer for the suggestion, and we removed the word "deeply" in Line 395 of the original manuscript. The revised text is shown below.

Section 3.1, Line 395: "*Therefore, in the Discussion section, we discuss the uncertainties of ensemble perturbations.*"

*6) Figure 4: It is unclear from the manuscript whether the experiments were run multiple times, particularly for plot d). The stochastic nature of this method may introduce variation in running time. Indicate whether the presented values are the mean of multiple runs or include the mean and uncertainty band for multiple runs.*

**Authors' response:**

We thank the reviewer for the suggestion. To test the dependence on ensemble size, we controlled a fixed assimilation frequency of 1 hour and changed the ensemble size to 2, 5, 10, 15, 20, 30, 40, and 50. Then we assimilated the hourly observations of five $PM_{2.5}$ chemical components from all sites with 48 timesteps from 00:00 (LST) on February $2^{nd}$ to 23:00 (LST) on February $3^{rd}$, 2022. This means that the Model Integration process and Assimilation process iteratively looped 48 times, which is equivalent to performing 48 experiments. Therefore, the presented values in Figure 4 are statistical results over 48 timesteps. For Fig. 4d, the elapsed time of the system processes are the statistical averages over 48 timesteps. According to the Reviewer's

suggestions, we indicated the presented values in Fig. 4 in the Line 381 of the original manuscript. The revised text is shown below.

Section 3.1, Line 381: "*Figure 4 shows the mean CRPS, RMSE and CORR values and the statistical averages of the elapsed time over 48 timesteps with the ensemble sizes of 2, 5, 10, 15, 20, 30, 40, and 50.*"

---

## Author Comment (AC2)

**Authors' responses to Referees' comments**

**Journal:** Geoscientific Model Development

**Manuscript Number:** gmd-2024-78

**Title:** NAQPMS-PDAF v2.0: A Novel Hybrid Nonlinear Data Assimilation System for Improved Simulation of $PM_{2.5}$ Chemical Components

**Authors:** Hongyi Li, Ting Yang, et al.

Note:

Comment (12-point black italicized font).

Reply (indented, 12-point blue normal font).

"Revised text as it appears in the text (in quotes, 12-point blue italicized font)".
* * *
**Anonymous Referee #2**

*1 General comments:*

*This paper builds on earlier developments in the NAQPMS-PDAF model. My main concern is that the writing reads very awkward in many places, please make sure the use of words is bringing what you wish to convey. Several sentences were conflicting or confusing because of ambiguous or inappropriate choice of words.*

**Authors' response:**

We thank the reviewer for the constructive suggestions in our manuscript. We apologize for some confusing aspects of this paper. We have thoroughly revised the entire paper based on your suggestions.

*2 Detailed Comments:*

*1) Line 13, 32: the first lines of the abstract and introduction are too similar.*

**Authors' response:**

We thank the reviewer for the valuable feedback. We reduce the similarity between the first lines of the Abstract and the Introduction. The revised text is shown below.

Abstract, Line 13: "*Identifying $PM_{2.5}$ chemical components is crucial for formulating emission strategies, estimating radiative forcing, and assessing human health effects.*"

Introduction, Line 32-34: "*$PM_{2.5}$ is a complex mixture of various chemical fractions,*

*mainly including sulfate ($SO_4^{2-}$), nitrate ($NO_3^-$), ammonium ($NH_4^+$), organic carbon (OC), and elemental carbon (EC). These chemical components exert diverse influences on the atmospheric environment (Khanna et al., 2018), human health (Bell et al., 2007; Schlesinger, 2007; Li et al., 2022a; Alves et al., 2023), and climate change (Schult et al., 1997; Park et al., 2014; Wilcox et al., 2016)."*

*2) Line 14: please note the difference between accuracy and precision… accuracy (accurately) fits the context more than precisely.*

**Authors' response:**

We thank the reviewer for the suggestion and agree that "accurately" fits the context more than "precisely". According to the reviewer's suggestion, we have replaced "precisely" with "accurately". The revised text is shown below.

Abstract, Line 14: *"However, accurately describing spatiotemporal variations of PM$_{2.5}$ chemical components remains a challenge."*

*3) Line 15: change difficulty to challenge, or a difficulty to challenging.*

**Authors' response:**

According to the reviewer's suggestion, we have replaced "difficulty" with "challenge". The revised text is shown below.

Abstract, Line 15: *"However, accurately describing spatiotemporal variations of PM$_{2.5}$ chemical components remains a challenge."*

*4) Line 33: delete diversely, it's unclear if the influence is diverse or the subjects are diverse.*

**Authors' response:**

We thank the reviewer for the suggestion, and we apologize for the lack of clarity in the original expression. The word "diverse" in Line 33 means that different PM$_{2.5}$ chemical components have different impacts on the environment, climate change, and human health.

For example, the dominant chemical components in haze events in different cities

in China vary significantly due to the presence of disparate emission sources (Huang et al., 2019). Regarding the influence on climate change, black carbon (BC) has positive radiative forcing, while sulfate ($SO_4^{2-}$) and nitrate ($NO_3^-$) have negative radiative forcing (Fig. R1). Anthropogenic $SO_4^{2-}$ increases summer precipitation, while absorptive BC has the opposite effect (Xie et al., 2022). Regarding the influence on human health, BC and $SO_4^{2-}$ have been shown to have a more significant pathogenic effect on dementia than $NO_3^-$ and OC in the northeastern United States (Li et al., 2022).

[Figure]

**Figure R1: Annual mean top of the atmosphere radiative forcing due to aerosol-radiation interactions ($RF_{ari}$, in W m$^{-2}$) due to different anthropogenic aerosol types, for the 1750–2010 period (quoted from Boucher et al. (2013)).**

The following revisions have been made to enhance the clarity of the presentation in the original manuscript.

Introduction, Line 32-34: "*PM$_{2.5}$ is a complex mixture of various chemical fractions, mainly including sulfate (SO$_4^{2-}$), nitrate (NO$_3^-$), ammonium (NH$_4^+$), organic carbon (OC), and elemental carbon (EC). These chemical components exert diverse influences on the atmospheric environment (Khanna et al., 2018), human health (Bell et al., 2007; Schlesinger, 2007; Li et al., 2022a; Alves et al., 2023), and climate change (Schult et al., 1997; Park et al., 2014; Wilcox et al., 2016).*"

**Reference**

Boucher, O., D. Randall, P. Artaxo, C. Bretherton, G. Feingold, P. Forster, V.-M. Kerminen, Y. Kondo, H. Liao, U. Lohmann, P. Rasch, S.K. Satheesh, S. Sherwood, B. Stevens and X.Y. Zhang: Clouds and Aerosols. In: Climate Change 2013: The Physical Science Basis. Contribution of Working Group I to the Fifth Assessment Report of the Intergovernmental Panel on Climate Change [Stocker, T.F., D. Qin, G.-K. Plattner, M. Tignor, S.K. Allen, J. Boschung, A. Nauels, Y. Xia, V. Bex and P.M. Midgley (eds.)]. Cambridge University Press, Cambridge, United Kingdom and New York, NY, USA, 2013

Huang, R., Zhang, Y., Bozzetti, C. et al.: High secondary aerosol contribution to particulate pollution during haze events in China. Nature, 514, 218–222, https://doi.org/10.1038/nature13774, 2014.

Li, J., Wang, Y., Steenland, K., et al.: Long-term effects of $PM_{2.5}$ components on incident dementia in the northeastern United States, Innovation, 3, 100208, https://doi.org/10.1016/j.xinn.2022.100208, 2022.

Xie, X., Myhre, G., Shindell, D., et al.: Anthropogenic sulfate aerosol pollution in South and East Asia induces increased summer precipitation over arid Central Asia. Commun. Earth Environ., 3, 328, https://doi.org/10.1038/s43247-022-00660-x, 2022.

*5) Line 37: "insufficient in interpreting PM2.5 chemical components" This is untrue or at least in literal text, this is claiming that measurements are not enough to interpret PM2.5 chemical components. However, observations have shown different aerosol composition (see AMS data). Please consider revising this claim. It doesn't seem to be what you want to say.*

**Authors' response:**

We thank the reviewer for the suggestion and agree that observations have shown different aerosol chemical compositions. The observation techniques for $PM_{2.5}$ chemical components can be classified into direct measurement techniques (such as aerosol mass spectrometer (AMS) and ion chromatography) and indirect observation techniques (such as ground-based or satellite-based remote sensing inversion).

Direct measurement techniques from research laboratories and field campaigns provide accurate and reliable data on the various $PM_{2.5}$ chemical components. However, these techniques are constrained by the limited number of sites or platforms and the high costs associated with measurements, which results in a lack of continuity in the

spatiotemporal dimension of the data obtained. Moreover, both ground-based remote sensing with a restricted number of sites and satellite-based remote sensing with temporal discontinuity in the fixed location are inadequate for interpreting the continuously spatiotemporal distribution of aerosol components. Furthermore, the uncertainty in the inversion algorithm limits the remote sensing data quality. Therefore, both direct and indirect observation techniques are insufficient for the acquisition of spatiotemporally continuous information of chemical components.

To obtain the spatiotemporally continuous information of chemical components, we developed a data assimilation system to generate gridded data with a spatial resolution of 5 km and a temporal resolution of 1 hour by combing a numerical model with the measurements from a limited number of dispersed observation sites.

The following revisions have been made to enhance the clarity of the presentation in the original manuscript.

Introduction, Line 35-38: "*However, current detection technologies, such as direct observation by sampling and chemical analysis (Zhang et al., 2015; Ming et al., 2017), ground-based remote-sensing inversion (Nishizawa et al., 2008; Nishizawa et al., 2011; Nishizawa et al., 2017), and observation-based machine learning (Lin et al., 2022; Su Lee et al., 2023; Li et al., 2025), are insufficient in interpreting spatiotemporally continuous information of PM$_{2.5}$ chemical components due to the limited number of observation sites or platforms.*"

*6) Line 43: What kind of "biases relative to real situation"? How?*

**Authors' response:**

In contrast to observations, atmospheric chemical transport models (CTMs) utilize computer-based techniques and numerical calculations to approximate the atmospheric state without using observational equipment or instruments. CTMs take meteorological data, emission inventory, and initial and boundary conditions as inputs and employ a series of physicochemical calculations to generate the spatiotemporally continuous simulation or forecast fields of PM$_{2.5}$ chemical components. The widely used CTMs include GEOS-Chem (Li et al., 2020), WRF-Chem (Lv et al., 2020), WRF-CAMx (Jia

et al., 2017), CMAQ (Wang et al., 2015; Yang et al., 2019) and NAQPMS (Wang et al., 2013).

However, there are notable discrepancies between the CTMs output and the observations. For instance, approximately 60% of the studies have indicated that the current models tend to overestimate the simulation of $NO_3^-$ concentration (Xie et al., 2022), which can be attributed to the deficiencies in the physicochemical mechanisms within the models (Zhang et al., 2015) and the impact of inadequate simulation of other chemical components (Song et al., 2021). Furthermore, CTMs are constrained by inaccuracies in the initial boundary conditions and uncertainties in the emission inventory and meteorological fields (Sax and Isakov, 2003; Rodriguez et al., 2007; Miao et al., 2020; Chang et al., 2015; Mallet and Sportisse, 2006; Huang et al., 2019), resulting in unreliable output from CTMs.

In this paper, biases are defined as the discrepancies between model simulations and accurate observations. These discrepancies are caused by uncertainties within the CTMs. The statement "*resulting in biases relative to real situation*" is inappropriate. The revised text is shown below.

[revised manuscript text omitted]

Zhang, R., Wang, G., Guo, S., Zamora, M. L., Ying, Q., Lin, Y., Wang, W., Hu, M., and Wang, Y.: Formation of urban fine particulate matter, Chem. Rev., 115, 3803-3855, https://doi.org/10.1021/acs.chemrev.5b00067, 2015.

*7) Line 91: Delete Besides or use another word.*

**Authors' response:**

According to the reviewer's suggestion, we delete "Besides". The revised text is shown below.

Introduction, Line 90-91: "*Section 3 presents the DA results, including an evaluation of dependencies, performance, and external comparisons, as well as a discussion of the ensemble DA uncertainty.*"

*8) Section 2.1: please expand the description for people unfamiliar with the model. This is a GMD paper after all.*

**Authors' response:**

We thank the reviewer for the suggestion and have expanded the description of NAQPMS in Section 2.1. The revised text is shown below.

Section 2.1, Line 95-101: "*The Nested Air Quality Prediction Modeling System (NAQPMS), developed by the Institute of Atmospheric Physics (IAP), Chinese Academy of Sciences (CAS), is used to provide background fields for key aerosol chemical components in this study. NAQPMS is a multi-scale gridded 3-dimensional Eulerian chemical transport model based on continuity equations. The nested grids in the horizontal direction enable data exchange between different domains. Applying terrain-*

*following coordinates in the vertical direction mitigates numerical calculation errors to enhance model accuracy. The NAQPMS comprises an input section, a numerical computation section, and an output section. The input section incorporates static terrain data, emission inventories, meteorological fields, and initial-boundary conditions. The numerical computation section performs multiple physicochemical process calculations, including the advection process, eddy diffusion, dry deposition, wet scavenging, gas-phase chemistry, aqueous chemistry, aerosol physicochemical processes (including heterogeneous reactions at the aerosol surface), and other processes. The schemes and features of the physicochemical processes are summarized in Table S1. The output section is responsible for model post-processing, data diagnostics, and source identification.*

*NAQPMS is capable of characterizing the three-dimensional spatiotemporal distribution of various atmospheric compositions at global and regional scales and has been widely used in atmospheric pollution and chemistry research, such as $O_3$ pollution (Wang et al., 2001), haze episodes (Wang et al., 2014; Du et al., 2021), regional transport (Wang et al., 2017; Wang et al., 2019), source identification (Li et al., 2022b), air quality simulation at global scale (Ye et al., 2021) and at urban-street scale (Wang et al., 2023), and acid deposition (Ge et al., 2014)."*

*9) Line 116: "PDAF has offline and online modes."*

**Authors' response:**

According to the reviewer's suggestion, the revised text is shown below.

Section 2.2, Line 116: "*PDAF has offline and online modes.*"

*10) Line 117: ", which is easy to write code" does not fit here.*

**Authors' response:**

We thank the reviewer for the suggestion, the revised text is shown below.

Section 2.2, Line 116-117: "*For the offline mode, PDAF and the model perform separately without coupling, obviating the need to modify the model code.*"

*11) Line 119: "instead of twice independently" What does this mean? what needs twice independently?*

**Authors' response:**

We apologize for the lack of clarity in the original manuscript. The statement "*instead of twice independently*" indicates that the initialization of the PDAF and the model need to be executed separately in the offline mode, resulting in the initialization process being executed twice. In contrast, the initialization of PDAF and the model is executed integrally in the online mode, with the initialization occurring only once. The revised text is shown below.

Section 2.2, Line 118-119: "*Firstly, the initialization of the PDAF and the model is integrated, necessitating a single execution rather than two separate executions.*"

*12) Figure 1: please write the description in the captions and discuss it in the text.*

**Authors' response:**

According to the reviewer's suggestion, the revised text is shown below.

Section 2.3.1, Line 138-143: "*Figure 1 illustrates the structure (left portion) and main workflow (right portion) of NAQPMS-PDAF v2.0. As described in the left portion of Fig. 1, the observation part involves the integration of multi-type observations (the purple cuboid patterns) and the utilization of PDAF-OMI. PDAF-OMI enables the simultaneous access and scheduling of multi-type and multi-source observations by employing observational indices, thereby facilitating flexible combinations of observations. The ensemble initial fields (the deep blue cuboid patterns) are crucial inputs for the numerical simulation of NAQPMS. The ensemble forecast/background fields (the deep yellow cuboid patterns) are generated by perturbing emission species based on hypothesized distributions (see Sect. 2.3.3) and performing physiochemical calculations in NAQPMS (the green rectangular patterns). Then, chemical DA is performed by a novel hybrid localized nonlinear DA algorithm (LKNETF, see Sect. 2.3.4) with an adaptive hybrid weight and an adaptive forgetting factor to generate analysis fields (the orange cuboid patterns) for the next realization.*"

[Figure]

*Figure 1: The structure of NAQPMS-PDAF v2.0 (Left: the purple cuboid patterns represent the multi-type observations, the deep blue cuboid patterns represent the initial fields, the deep yellow cuboid patterns represent the forecast or background fields, and the orange cuboid patterns represent the analysis fields. Ens.1st represents the first ensemble member, and Ens.Nth represents the Nth ensemble member. Right: the main workflow in NAQPMS-PDAF v2.0, blue rectangular patterns represent the modules in NAQPMS, and yellow rectangular patterns represent the modules in PDAF).*"

Section 2.3.1, Line 146-154: "*NAQPMS-PDAF v2.0 implements an online coupling between NAQPMS and PDAF v2.1 with OMI, utilizing a level-2 parallel computational framework. The description of level-2 parallel implementation was detailed in our previous work (Wang et al., 2022). The online coupling ensures the continuous operation of model forecasts and assimilation analysis at each time step, achieved by directly integrating PDAF routines into the prototype code of NAQPMS (the right portion of Fig. 1, the blue represents NAQPMS main routines, while the yellow represents PDAF main routines). The level-2 parallel computational framework, which utilizes the Message Passing Interface standard (MPI), facilitates concurrent processing and data exchange among multiple ensemble members and parallel*

*computation among model state matrixes within each ensemble member, enhancing the efficiency of ensemble analysis and numerical model computations. For instance, the operation of twenty ensemble members necessitates the execution of twenty model tasks, each of which performs integral computation on a large model grid. Twenty model tasks can be executed simultaneously at twenty computational nodes with sufficient computational resources. Each model task can then perform parallel computation with multiple processors by splitting the large model grid into multiple sub-grids. As illustrated in the right portion of Fig. 1, the workflow of NAQPMS-PDAF v2.0 is outlined as follows:*"

*13) Line 153-171: consider using a flow chart to illustrate the steps.*

**Authors' response:**

We thank the reviewer for the suggestion. The flowchart was shown in the right portion of Fig. 1. The steps are specific explanations of the workflow of NAQPMS-PDAF v2.0. We have incorporated the requisite figure citation into the text.

Section 2.3.1, Line 153-154: "*As illustrated in the right portion of Fig. 1, the workflow of NAQPMS-PDAF v2.0 is outlined as follows:*"

*14) Line 172: Configurations*

**Authors' response:**

We thank the reviewer for the suggestion. The revised text is shown below.

Section 2.3.2, Line 172: "*2.3.2 Configurations*"

*15) Line 217-218: The target PM2.5 chemical components are NH4 + , SO4 2- , NO3 - 217 , OC, and EC, and the perturbed emission species correspondingly 218 include SO2, NOx, VOCs, NH3, CO, PM10, PM2.5, EC, and OC,'' You have a target of 5, and corresponding to 9 species… please be specific.*

**Authors' response:**

We thank the reviewer for the insightful comment and apologize for the confusion caused by the initial description.

Ensemble data assimilation employs multiple forecast ensemble members to represent model uncertainty and calculate background error covariances. Emission inventory input represents a significant source of uncertainty in atmospheric chemistry transport models (CTMs). We generate a set of forecast ensembles by perturbing the emission inventory input from Multi-resolution Emission Inventory for China (MEIC) data. The emission inventory input contains 25 emission species, of which 9 emission species closely related to the formation and evolution of the five $PM_{2.5}$ chemical fractions were selected for perturbation. The 9 emission species are all involved in the calculation of gas-phase chemical process (CBM-Z, Zaveri and Peters, 1999) and inorganic aerosol process (ISORROPIA, Nenes et al., 1998) in the model, which directly or indirectly affect the $PM_{2.5}$ chemical component concentrations. For example, sulfur dioxide ($SO_2$), nitrogen oxides ($NO_x$), and ammonia ($NH_3$) are significant precursor gases for sulfate ($SO_4^{2-}$), nitrate ($NO_3^-$), and ammonium ($NH_4^+$), respectively (Geng et al., 2017). The emissions of organic carbon (OC) and elemental carbon (EC) directly impact the calculation of OC and EC concentrations, while volatile organic compounds ($VOC_s$) directly influence the formation of secondary organic aerosol (SOA) (Miao et al., 2020; Chen et al., 2022). Carbon monoxide (CO) indirectly influences the calculation of component concentrations by participating in the calculation of inorganic gas-phase chemistry process (Zaveri and Peters, 1999) and the estimation of SOA precursor surrogates in CTMs (Miao et al., 2020). The emission of $PM_{2.5}$ and $PM_{10}$ can provide particle surfaces for the adsorption of gases, thereby facilitating heterogeneous reactions that result in the formation of secondary pollutants (Zhu et al., 2011).

According to the Reviewer's suggestion, the revised text is shown below.

Section 2.3.3, Line 217-218: "*The target $PM_{2.5}$ chemical components are $NH_4^+$, $SO_4^{2-}$, $NO_3^-$, OC, and EC. The perturbed emission species that can directly or indirectly affect the component concentration calculations include $SO_2$, $NO_x$, $VOC_s$, $NH_3$, CO, $PM_{10}$, $PM_{2.5}$, EC, and OC, with the corresponding uncertainties (δ) listed in Table 1.*"

**Authors' response:**

The term "superiority" is used to evaluate the competitiveness of NAQPMS-PDAF v2.0 in generating the reanalysis datasets of the $PM_{2.5}$ chemical compositions in comparison to the other reanalysis datasets, including CAMSRA and MERRA-2. The assessment of data superiority includes (1) the ability to characterize the spatial distribution of $PM_{2.5}$ chemical components and (2) the accuracy of the concentration values.

In the first assessment, the spatial distributions of monthly mean concentrations from ground-based observations at 33 sites, the analysis field dataset of NAQPMS-PDAF v2.0, and other reanalysis datasets were compared. In the second assessment, the statistical indicators, including the correlation coefficient, RMSE, and $R^2$, were quantified between the datasets and observations. The results of the two assessments allowed us to compare the data quality between the NAQPMS-PDAF v2.0 analysis field dataset and other reanalysis datasets.

To clarify the meaning of the term "*superiority*", we have revised the text at Line

343 in the original manuscript as follows.

 "*...the quality of output data compared to other reanalysis datasets...*"

*17) Line 352: situation ->scenario or test*

**Authors' response:**

According to the reviewer's suggestion, the term "situation" was replaced with "scenario". Please find the revised text below.

Section 2.5, Line 350-353: "*In the first scenario, we controlled a fixed assimilation frequency of 1 hour and changed the ensemble size to 2, 5, 10, 15, 20, 30, 40, and 50. In the second scenario, we controlled a fixed ensemble size of 20 and changed the assimilation frequency to 1 hour, 2 hours, 3 hours, 4 hours, 5 hours, 6 hours, 8 hours, and 12 hours.*"

Section 2.5, Line 359-360: "*From Table 2, we fixed species uncertainty (M4 setting) with five distribution types in the first scenario and fixed distribution type (T2 setting) with five $SO_2$ uncertainties in the second.*"

*18) Line 358: "The last test was like the first but with a different situation" This is too colloquial*

**Authors' response:**

We thank the reviewer for the suggestion. The revised text is shown below.

Section 2.5, Line 358-359: "*The final test was analogous to the first test but with a distinct scenario designed to examine the influence of ensemble perturbation on ensemble assimilation.*"

*19) Figures 6-10: Figures are unreadable. Too small. Please re-plot.*

**Authors' response:**

We thank the reviewer for the valuable feedback. We apologize for the inconvenience caused by the unreadable figures. We have revised Figures 6-10 to make

the font larger and clearer, and the figures more legible. Please find the revised figures below.

[Figure]

*Figure 6: Scatterplots of the DA-site simulations versus the DA-site observations with probability density for the free-running field (FR, a1-a5), forecast field (FOR, b1-b5), and analysis field (ANA, c1-c5). The dotted gray lines represent the 2:1, 1:1, and 1:2 lines, and the solid red line represents the fitting regression line.*

[Figure]

*Figure 7: Probability distributions of bias between DA-site observations and DA-site simulations for the free-running field (FR, a1-a5), forecast field (FOR, b1-b5), and analysis field (ANA, c1-c5).*

[Figure]

*Figure 8: Hourly variation of five PM₂.₅ chemical components in a representative DA site (a) and a representative VE site (b).*

[Figure]

*Figure 9: Hourly variation of PM₂.₅ in three representative DA sites (a1-a3) and three representative VE sites (b1-b3).*

[Figure]

*Figure 10: Spatial concentration distribution of site observation (OBS, a1-e1), free-run field (FR, a2-e2), forecast field (FOR, a3-e3), analysis field (ANA, a4-e4), and increment (INC) between ANA and FR (a5-e5) for five PM₂.₅ chemical components.*

[Figure]

*Figure S3: Scatterplots of the VE-site simulations versus the VE-site observations with probability density for the free-running field (FR, a1-a5), forecast field (FOR, b1-b5), and*

*analysis field (ANA, c1-c5). The dotted gray lines represent the 2:1, 1:1, and 1:2 lines, and the solid red line represents the fitting regression line.*

[Figure]

*Figure S4: Probability distributions of bias between VE-site observations and VE-site simulations for the free-running field (FR, a1-a5), forecast field (FOR, b1-b5), and analysis field (ANA, c1-c5).*